# Ten-eleven translocation 1 mediated-DNA hydroxymethylation is required for myelination and remyelination in the mouse brain

Ming Zhang [1,8], Jian Wang[1,8], Kaixiang Zhang[1,8], Guozhen Lu[1], Yuming Liu[1], Keke Ren[1], Wenting Wang [1], Dazhuan Xin[2], Lingli Xu[3], Honghui Mao[1], Junlin Xing[1], Xingchun Gao[4], Weilin Jin[5], Kalen Berry [2], Katsuhiko Mikoshiba [6,7], Shengxi Wu [1✉], Q. Richard Lu [2✉] & Xianghui Zhao [1✉]

Ten-eleven translocation (TET) proteins, the dioxygenase for DNA hydroxymethylation, are important players in nervous system development and diseases. However, their role in myelination and remyelination after injury remains elusive. Here, we identify a genome-wide and locus-specific DNA hydroxymethylation landscape shift during differentiation of oligodendrocyte-progenitor cells (OPC). Ablation of *Tet1* results in stage-dependent defects in oligodendrocyte (OL) development and myelination in the mouse brain. The mice lacking *Tet1* in the oligodendrocyte lineage develop behavioral deficiency. We also show that TET1 is required for remyelination in adulthood. Transcriptomic, genomic occupancy, and 5-hydroxymethylcytosine (5hmC) profiling reveal a critical TET1-regulated epigenetic program for oligodendrocyte differentiation that includes genes associated with myelination, cell division, and calcium transport. *Tet1*-deficient OPCs exhibit reduced calcium activity, increasing calcium activity rescues the differentiation defects in vitro. Deletion of a TET1-5hmC target gene, *Itpr2*, impairs the onset of OPC differentiation. Together, our results suggest that stage-specific TET1-mediated epigenetic programming and intracellular signaling are important for proper myelination and remyelination in mice.

[1] Department of Neurobiology, School of Basic Medicine, Fourth Military Medical University, Xi'an, Shaanxi, China. [2] Division of Experimental Hematology and Cancer Biology, Brain Tumor Center, Cincinnati Children's Hospital Medical Center, Cincinnati, OH, USA. [3] Center for molecular medicine, Pediatric Research Institute, Children's Hospital of Fudan University, Shanghai, China. [4] Shaanxi Key Laboratory of Brain Disorders, Xi'an Medical University, Xi'an, Shaanxi, China. [5] Institute of Cancer Neuroscience, Medical Frontier Innovation Research Center, The First Hospital of Lanzhou University, Lanzhou, China. [6] Faculty of Science, Toho University, Funabashi, Japan. [7] Present address: Shanghai Institute for Advanced Immunochemical Studies, ShanghaiTech University, Shanghai, China. [8] These authors contributed equally: Ming Zhang, Jian Wang, Kaixiang Zhang. ✉email: shengxi@fmmu.edu.cn; richard.lu@cchmc.org; xianghuizhao@fmmu.edu.cn

Myelination by oligodendrocytes (OLs) enables saltatory conduction of action potentials and provides long-term trophic support for axons, maintaining integrity throughout the central nervous system (CNS)[1]. The formation of mature myelinating OLs is a complex process that is tightly coordinated spatially and temporally by genetic and epigenetic events[2,3]. Epigenetic regulation by DNA methylation, histone modification, and chromatin remodeling is critical for multiple aspects of OL development, function, and regeneration[4–6]. For instance, proper maintenance of genomic 5-methylcytosine (5mC) is essential for normal development, homeostasis, and function of mammalian cells[7,8]. Genetic ablation of *Dnmt1*, which encodes the DNA methyltransferase that maintains DNA methylation after replication, results in impaired OL precursor cell (OPC) expansion and differentiation during early development[9].

The modified nucleotide 5-hydroxymethylcytosine (5hmC) has been shown to be an intermediate product generated during cytosine demethylation[10,11]. DNA demethylation, like methylation, is a highly regulated process. DNA demethylation is mediated by the ten-eleven translocation (TET) family of dioxygenases. The TET enzymes oxidize 5mC into 5hmC to initiate the DNA demethylation process[11,12]. Dynamic regulation of cytosine methylation or demethylation has been established as a common epigenetic modification regulating various processes from development to diseases in a cell-type and context-dependent manner[13–15]. TET enzymes are present in OL lineage cells[16], and here we interrogated how DNA demethylation contributes to OL lineage development, myelination, and remyelination after injury.

In this study, we demonstrate that there is a genome-wide shift in the 5hmC landscape during OL specification and identify an age-dependent function of TET1 in OL lineage specification and myelination. The mice with *Tet1* deletion in OL lineage develop behavioral deficiency. In addition, we show that a TET1-regulated epigenetic program is required for efficient remyelination as depletion of *Tet1* in OPCs impairs myelin recovery after demyelinating injury in adult animals. Moreover, *Tet1* depletion resulted in genome-wide alterations in 5hmC and transcriptomic profiles that are associated with OPC differentiation and myelination, as well as calcium transport. Ablation of *Itpr2*, one of the TET1-5hmC targets that responsible for calcium release from endoplasmic reticulum (ER) in the OL lineage significantly impairs OL differentiation. These data suggest that TET1 and DNA hydroxymethylation mediated transcriptional and epigenetic programming regulate OL intracellular signaling and are required for proper myelination and animal behaviors.

## Results

### Dynamic DNA hydroxymethylation landscape during OL lineage specification

To investigate the 5hmC landscape during OL lineage specification, we carried out antibody-based 5hmC immunoprecipitation (IP) combined with Illumina sequencing (hMeDIP-seq)[17,18] and analyzed 5hmC distribution across the genome. We compared the 5hmC distribution within OPCs isolated via immunopanning from the cortices at postnatal day 6 (P6) to that in neural progenitor cells (NPCs)[19] and identified 1237 genes that were specifically hydroxymethylated in the promoter or transcription start site (TSS) regions of OPCs but not NPCs (Fig. 1a). Gene ontology (GO) analysis revealed that these genes involved in OPC differentiation are highly associated with terms such as cell projection organization, fatty acid transport, and regulation of cytosolic calcium ion concentration, and with signaling pathways that are essential for OL development such as the G-protein coupled receptor pathway[20,21] (Fig. 1b). Similarly,

gene set enrichment analysis (GSEA) for 5hmC peaks in the gene body regions indicated that genes associated with the bipotent progenitor, OL progenitor, and postmitotic OL were enriched in OPCs (Fig. 1c), while pluripotent stem cell-associated genes were enriched in NPCs (Fig. 1c). Comparison with a neural cell-type transcriptome dataset[22] (Supplementary Fig. 1a–c) showed that the 5hmC signals were higher in OPCs than NPCs, in gene loci of OPC-associated genes, e.g., *Cspg4* (chondroitin sulfate proteoglycan 4) (Fig. 1d), immature OL-associated genes, e.g., *Kndc1* (kinase non-catalytic C-lobe domain containing 1) (Fig. 1e), and mature OL-associated genes, e.g., *Mag* (myelin-associated glycoprotein) (Fig. 1f). In contrast, the genes with 5hmC peaks enriched in NPCs were associated with negative regulation of OPC differentiation, such as *Ngf* and *Zfp28* (Fig. 1g). We further verified the presence of loci-specific hydroxylmethylation in representative genes with a qPCR assay based on a combination of bisulfite and subsequent cytosine deaminase (APOBEC) treatment (Supplementary Fig. 1d and Supplementary Table 1). These data suggested a unique distribution pattern of genomic 5hmCs in the gene loci associated with OL lineage specification during the transition from NPCs to OPCs.

### Deletion of *Tet1* in OL lineage causes myelination deficits at early postnatal stages

TET1-3 enzymes are present in OL lineage cells[16]. As TET2 had no detectable effects in OL lineage development[23], we assessed the functions of TET1 and TET3 in OL development. We crossed *Tet1*^flox/flox^ mice[24] and *Tet3*^flox/flox^ mice[18] with the *Olig1*-Cre line[25] to knockout the catalytic domains of these TET enzymes early in OL lineage development (Fig. 2a and Supplementary Fig. 2a). The resulting *Tet1*^flox/flox^;*Olig1*Cre^+/−^ (*Tet1* cKO) and *Tet3*^flox/flox^;*Olig1*Cre^+/−^ (*Tet3* cKO) mice were born at Mendelian ratios and appeared normal at birth. We did not detect significant differences in either the number of CC1^+^ mature OLs or myelin protein expression between heterozygous *Tet1*-floxed mice (*Tet1*^flox/+^;*Olig1*Cre^+/−^), *Cre* control (*Tet1*^+/+^;*Olig1*Cre^+/−^), or wild-type mice (Supplementary Fig. 3a, b). Also, we did not detect any OLIG2 expression changes within the *Olig1*-Cre (+/−) heterozygous line (Supplementary Fig. 3c, d). Therefore, heterozygous littermates were used as controls. To assess Cre-mediated *Tet1* depletion in OL lineage, we quantified TET1 expression in OPCs from *Tet1* cKO and control mice at P4. Immunostaining revealed that expression of TET1 in SOX10^+^ OLs was significantly reduced in *Tet1* cKO than in control mice (Fig. 2b, c). TET1 levels were also decreased in purified OPCs from *Tet1* mutant than from control mice assayed by quantitative real-time PCR (Supplementary Fig. 3e). To examine the cell-type or lineage-specificity of *Olig1*-Cre in the postnatal brain, we have generated *Olig1*-Cre; R26-tdTomato mice by crossing the *Olig1*-Cre mice with R26-tdTomato reporter line (Ai14). We found that most *Olig1*-Cre-TdTomato cells were OLIG2-positive OL lineage cells (Supplementary Fig. 3f, g), while TdTomato^+^ cells rarely co-label with the markers for astrocytes (ALDH1L1^+^), neurons (NeuN^+^), or interneurons (PV^+^, SST^+^, and VIP^+^) (Supplementary Fig. 3f, h, i), indicating that the *Olig1*-Cre line is predominantly restricted to the OL lineage.

To investigate the effects of TET1 on OL development in the brain, we examined the expression of OL lineage marker SOX10 and mature OL markers CC1 and MBP. The number of CC1^+^ mature OLs was significantly reduced in juvenile *Tet1* cKO mice compared to controls (Fig. 2d, e), but this difference was not observed in adult mice from P52 (Fig. 2e and Supplementary Fig. 4a). Expression of MBP was also substantially decreased in both cortex (gray matter) and corpus callosum (white matter) in *Tet1* cKO mice compared to controls at P16 (Fig. 2f), but the levels were similar in P60 adult animals (Supplementary Fig. 4b).

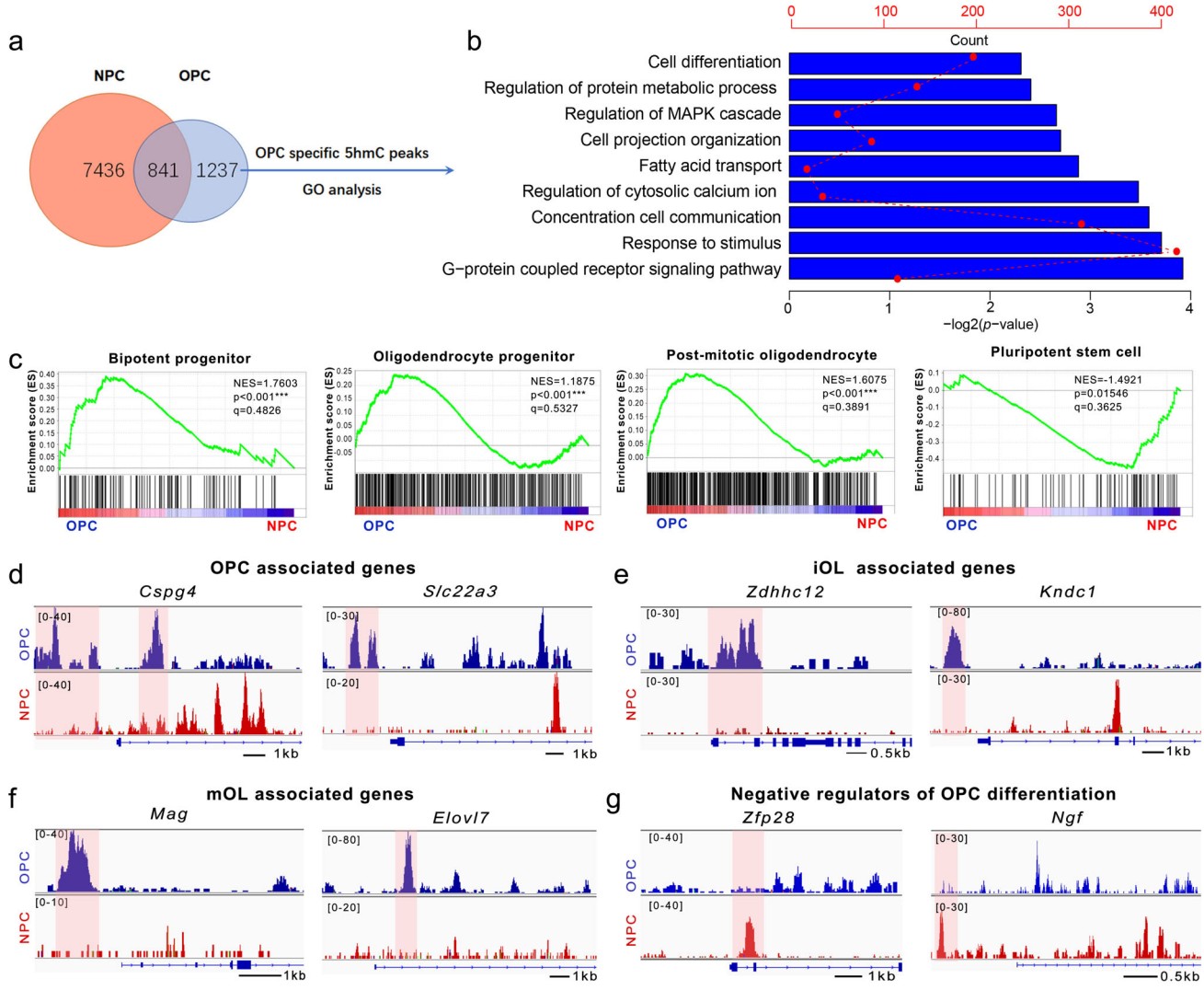

**Fig. 1 Dynamic 5hmC expression pattern during OPC specification. a** Venn diagram of hydroxymethylated genes in NPCs (neural progenitor cells) and OPCs (oligodendrocyte progenitor cells). **b** Gene ontology (GO) analysis of genes with OPC-specific 5hmC peaks in promoter or TSS regions. **c** GSEA plots of gene terms involved in "bipotent progenitor", "oligodendrocyte progenitor", "postmitotic oligodendrocyte", and "pluripotent stem cell" for genes with 5hmC peaks in gene body region from NPCs and OPCs. **d** Snapshots of 5hmC profiles of representative OPC-associated genes, *Cspg4* and *Slc22a3*, in NPCs and OPCs. Tracks were shown in IGV 2.4.8. **e** Snapshots of 5hmC profiles of representative immature oligodendrocyte (iOL)-associated genes, *Zdhhc12* and *Kndc1*, in NPCs and OPCs. **f** Snapshots of 5hmC profiles of representative mature oligodendrocyte (mOL)-associated genes, *Mag* and *Elovl7*, in NPCs and OPCs. **g** Snapshots of 5hmC profiles of representative negative regulators of oligodendrocyte differentiation, *Ngf* and *Zfp28* in NPCs and OPCs.

These data are consistent with previous observations with a siRNA-mediated knockdown in rat OPC culture[16] and indicate that *Tet1* loss causes a delay in OL maturation. Similar experiments in the *Tet3* cKO animals did not show any significant differences between mutants and controls (Supplementary Fig. 2), which is different from previous in vitro data using siRNA-mediated knockdown[16]. We speculate that this might be due to the possible off-target effects of TET3 siRNA on other TET members or other unidentified factors crucial for OPC differentiation. Therefore, we focused on examining the processes underlying the observed myelination defects in *Tet1* cKO mice.

In addition, electron microscopy (EM) revealed that the number of myelinated axons was significantly reduced in *Tet1* mutants compared to controls at both P14 optic nerves and P27 corpus callosum (Fig. 2g, h, j). Moreover, those myelinated axons in *Tet1* cKO mice were characterized by higher G ratios and thinner myelin sheaths than those of control mice (Fig. 2i, k). However, the myelin ultrastructure defects were not observed in P60 adult *Tet1* cKO animals (Supplementary Fig. 4c–e). Together, these results suggest a stage-dependent function of TET1 in CNS myelination.

Consistent with the observation, we confirmed here that TET1 expression significantly decreased in control adult OLs (Supplementary Fig. 4f, g). However, in *Tet1* cKO mice at P60, the number of TET1$^+$ OLs increased to ~29% (Supplementary Fig. 4f, g), which is higher than ~12% in *Tet1* cKO animals at P4 (Fig. 2b, c). This result indicates that OPCs failed to disrupt TET1 expression may contribute to the recovery of myelin formation in older mutant animals.

To evaluate the neurological significance of hypomyelination in *Tet1* cKO mice, we analyzed stimulus-evoked compound action potential (CAP) in P14 optic nerves as previously described[26,27]. Suction electrodes backfilled with artificial cerebrospinal fluid (aCSF) were used for stimulation and recording. In *Tet1* mutants, both the peak amplitudes and the CAP areas, which are indexes of excited myelinated axon numbers and nerve function[26,27], were significantly lower than controls under all stimulating currents tested (Fig. 2l–n). This observation indicates that

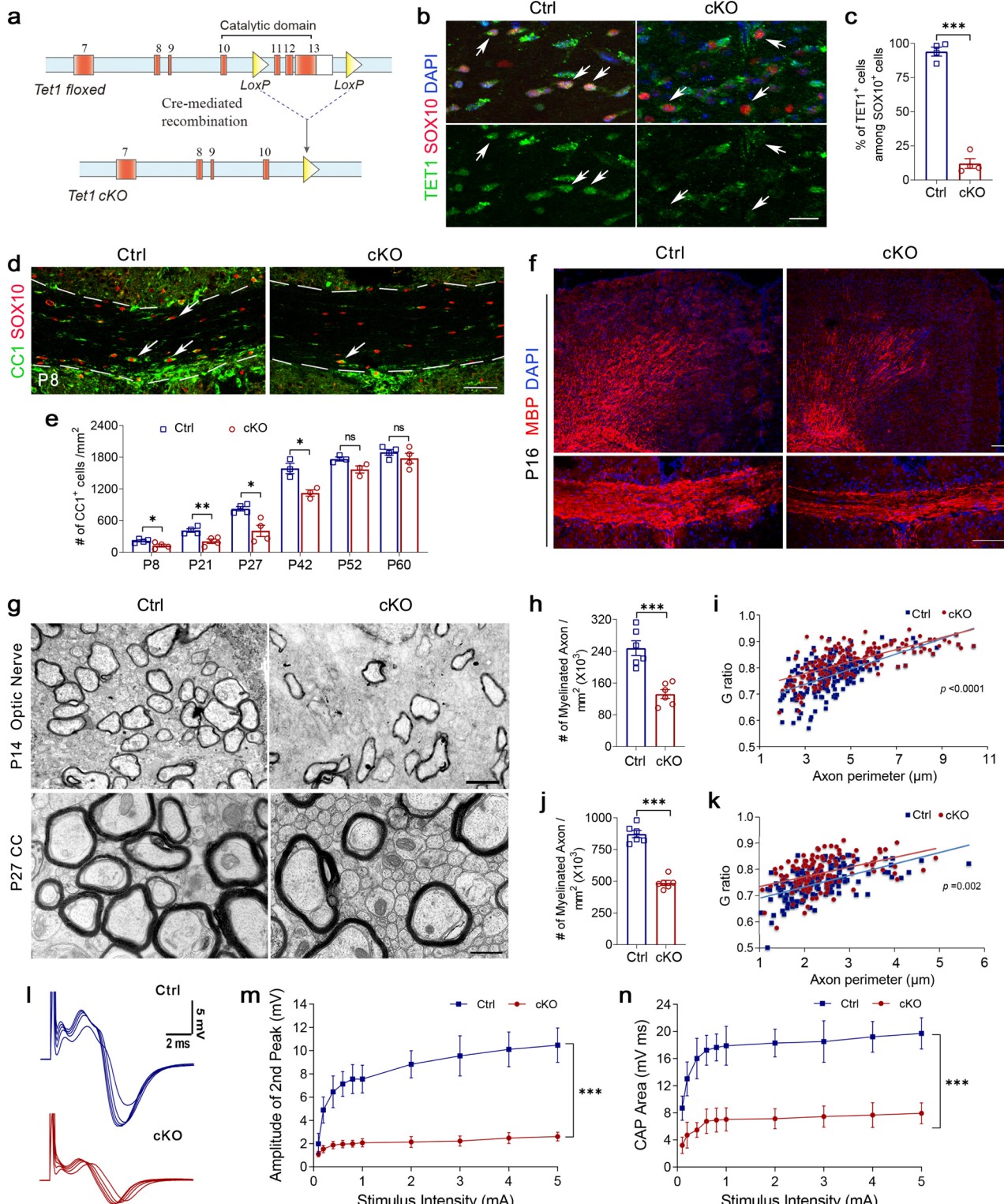

hypomyelination impairs action potential transduction in *Tet1* cKO mice.

**Loss of TET1 function in OL lineage cells induces behavioral deficiency in the mice.** Multiple studies have associate TET-5hmC with psychiatric and cognitive disorders[28–30], and multivariable logistic regression showed that ERBB4, BDNF, and TET1 were independent predictors for schizophrenia[31]. To gain insight into the physiological function of TET1 in animal behaviors, *Tet1*

cKO mice were subjected to behavior tests relevant to schizophrenia. *Tet1* cKO mice did not exhibit differences in weight and whisker number in comparison with control littermates.

First, we investigated the performance of juvenile *Tet1* mutant in prepulse inhibition (PPI) of the startle test at P42 with myelin deficiency (Supplementary Fig. 5a), which is a common test of sensorimotor gating ability for schizophrenia[32]. Reduced PPI ability due to an exaggerated acoustic startle reflex (ASR) is thought to contribute to schizophrenic conditions. We found that the

**Fig. 2 TET1 is required for OL differentiation and myelination. a** Schematic diagram of Cre-mediated excision of floxed *Tet1* exons encoding the critical catalytic domain for dioxygenase activity. **b** Representative immunostaining for TET1 expression in postnatal day 4 (P4) corpus callosum from control (Ctrl) and *Tet1* cKO (cKO) mice. Upper images show TET1 (green), SOX10 (red), and DAPI (blue) staining. Lower images show only TET1. Arrows indicate SOX10+ cells. Scale bar, 20 μm. **c** Percentage of TET1+ cells among SOX10+ oligodendrocytes in P4 corpus callosum of control and *Tet1* cKO mice. Data were Means ± SEM ($n = 4$ animals in each group). ***, Two-tailed unpaired *t*-test, $t = 17.091$, $df = 6$, $p = 0.000003$. **d** Representative images of CC1 and SOX10 immunostaining in the corpus callosum of control and *Tet1* cKO mice at P8. Scale bar, 50 μm. **e** Quantification of CC1+ cells in P8, P21, P27, P42, P52, and P60 control and *Tet1* cKO corpus callosum. Data were Means ± SEM ($n = 3$ animals in P42 and P52 for each group, $n = 4$ animals in P8, P21, P27, and P60 for each group). Two-tailed unpaired *t*-test, *, P8: $t = 2.724$, $df = 6$, $p = 0.034$; **, P21: $t = 3.794$, $df = 6$, $p = 0.009$; *, P27: $t = 3.702$, $df = 6$, $p = 0.010$; *, P42: $t = 4.071$, $df = 4$, $p = 0.015$; P52: $t = 2.529$, $df = 4$, $p = 0.065$; P60: $t = 1.036$, $df = 6$, $p = 0.340$ compared to control. **f** Representative images of MBP immunostaining in *Tet1* cKO and control mice cortex (upper images) and corpus callosum (lower images) at P16. Scale bar, 100 μm. **g** Representative electron micrographs of P14 optic nerve and P27 corpus callosum (cc) from control and *Tet1* cKO mice. Scale bar, 0.5 μm in P14 and 2 μm in P27, respectively. **h** Quantification of the number of myelinated axons in defined areas from the optic nerve of P14 control and *Tet1* cKO mice. Data were Means ± SEM ($n = 6$ slides from three animals per group). ***, Two-tailed unpaired *t*-test, $t = 5.308$, $df = 10$, $p = 0.000344$ compared to control. **i** G ratios vs. axonal perimeters for P14 optic nerve in control and *Tet1* cKO mice. Friedman M test, $\chi^2 = 43.251$, $df = 1$, $p_{between\ group} < 0.0001$ (>150 myelinating axon counts/animal from three animals/genotype). **j** Quantification of the number of myelinated axons in defined areas from corpus callosum of P27 control and *Tet1* cKO mice. Data were Means ± SEM ($n = 6$ slides from three animals per group). ***, Two-tailed unpaired *t*-test, $t = 11.050$, $df = 10$, $p < 0.0001$ compared to control. **k** G ratios vs. axonal perimeters for P27 corpus callosum in control and *Tet1* cKO mice. Two-way ANOVA, $F_{between\ group\ (1,58.597)} = 10.806$, $F_{within\ group\ (3,3)} = 71.262$, $p_{between\ group} = 0.002$ (>120 myelinating axon counts/animal from three animals/genotype). **l** Representative compound action potential (CAP) series from optic nerves of P14 control and *Tet1* cKO mice elicited by square voltage pulses of varied amplitudes. **m** Evoked CAP amplitudes of the second peak (maximal) from control and *Tet1* cKO mice plotted vs. stimulus intensities. Data were Means ± SD ($n = 11$ nerves per group). Two-way ANOVA, $F_{within\ group\ (9,200)} = 55.07$, $p < 0.0001$, ***, $F_{between\ group\ (1,200)} = 1804$, $p < 0.0001$. **n** Total CAP area vs. stimulus intensities in controls and *Tet1* cKO mice at all stimulus intensities compared. Data were Means ± SD ($n = 11$ nerves per group). Two-way ANOVA, $F_{within\ group\ (9,200)} = 30.31$, $p < 0.0001$, ***, $F_{between\ group\ (1,200)} = 1337$, $p < 0.0001$.

input/output function and the startle response were comparable between control and *Tet1*-mutant mice (Supplementary Fig. 5b), indicating the normal hearing and motor abilities (i/o function) in *Tet1* cKO mice. However, when using a combination of auditory-evoked startle (120 dB) and three levels of prepulse (70, 76, and 82 dB) to compare ARS, we observed that PPI was significantly attenuated in *Tet1* cKO mice in comparison with control animals (Supplementary Fig. 5c), suggesting the impaired sensorimotor gating ability in *Tet1* mutant.

Since working memory deficits are characteristic features in schizophrenia, adult *Tet1* cKO mice and control littermates at P90 were evaluated for their performance in the Morris water maze (MWM) to assess their working memory. Consistently, the myelin deficiency in *Tet1* cKO mice was recovered after P60 (Supplementary Fig. 4), including the hippocampus that is associated with spatial reference memory[33] (Supplementary Fig. 5d). Five-day acquisition trials exhibited similar swim paths, swim velocity, and escape latency to the platform between control and *Tet1* cKO groups (Supplementary Fig. 5e–g), which indicates that adult *Tet1* mutants had normal swimming and learning abilities. However, in the sixth-day probe trial, the escape latency was significantly higher in *Tet1* cKO mice than in control mice (Supplementary Fig. 5h), and the number of crossing the position was greatly reduced in mutant mice (Supplementary Fig. 5i). Additionally, in contrast to controls, *Tet1* cKO mice showed no preference for the target quadrant over other three quadrants (Supplementary Fig. 5j, k). These observations indicate that developmental myelinogenesis may contribute to working memory deficits in *Tet1*-mutant animals. Together, our results suggest that the abnormal myelination in *Tet1* cKO mice may cause behavioral deficiency with impaired sensorimotor gating ability and working memory.

**Ablation of *Tet1* results in defects in OPC cell-cycle progression.** Concomitant with the myelin deficiency, we observed a marked reduction of OLIG2+ cells from embryonic stage E15.5 and at P1 in *Tet1* cKO cortex relative to controls (Fig. 3a, b). Moreover, the number of PDGFRα+ OPCs in the mutant cortex was significantly reduced at E15.5 and P1 (Fig. 3a, c), suggesting a downsized OPC pool in *Tet1* cKO brain. Interestingly, the difference of OPC number between *Tet1* mutant and control brains was not obvious after P6 (Fig. 3c). We speculate that this may in

part be due to the substantial decline in TET1 expression with age particularly after P4[16], suggesting a less critical role for TET1 in later stages of OPC development. In addition, it is possible that compensation from other TETs (e.g., TET2 and TET3) might also contribute to OPC development in the *Tet1* cKO mice.

To determine the underlying defects that led to the observed reduction in the OPC and OL population in juvenile *Tet1* mutants, we first tested the possibility that OPCs are more likely to undergo apoptosis in the mutant with the TUNEL assay. Brain sections from E14.5, E17.5, and P1 mice revealed no distinguishable changes in the number of apoptotic cells among OLIG2+ OL lineage cells between *Tet1* cKO animals and control littermates (Supplementary Fig. 6a, b).

Next, we performed a BrdU incorporation assay to examine the proliferation of OPCs. At P1, mice were dosed with BrdU and sacrificed 2 h later. Compared to controls, intriguingly, the percentage of BrdU+ cells in OLIG2+ OL lineage cells showed a significant increase in *Tet1* cKO cortex (Fig. 3d, e). However, BrdU incorporation in older mice, e.g., P20 and P60, did not reveal the difference in OPCs between groups (Supplementary Fig. 6c–f). In addition, we examined the proliferation changes in purified OPC cultures. Both BrdU incorporation and Ki67 staining showed increased proliferating OPCs from *Tet1* mutants (Supplementary Fig. 6g–i).

The reduction of OPC numbers in *Tet1* cKO mice thus promote us to investigate if there is a cell-cycle defect in *Tet1*-deficient OPCs. Then we performed flow cytometry for purified OPCs in which DNA was stained with propidium iodide. Significant increases in the percentages of cells in the S phase (23.49 ± 0.85%) and G2/M phase (13.39 ± 1.01%) were observed in OPCs from *Tet1* mutants compared to the controls (9.02 ± 0.99% for S phase and 8.29 ± 0.62% for G2/M phase) (Fig. 3f, g and Supplementary Fig. 7a, b). Moreover, there was a concomitant reduction in the number of cells in the G1 phase in *Tet1* mutants compared to the controls (Fig. 3f, g).

A recent report has suggested the critical role of TETs in regulating the G2 to M phase progression of trophoblast stem cells[34], we then tested G2-M progression of *Tet1* cKO OPCs in a cell cycle synchronization assay. We arrested cell cycle at G2/M with the CDK1 inhibitor RO3306 for 12 h, followed by a release at specific time points (Fig. 3h). A marker for the M phase,

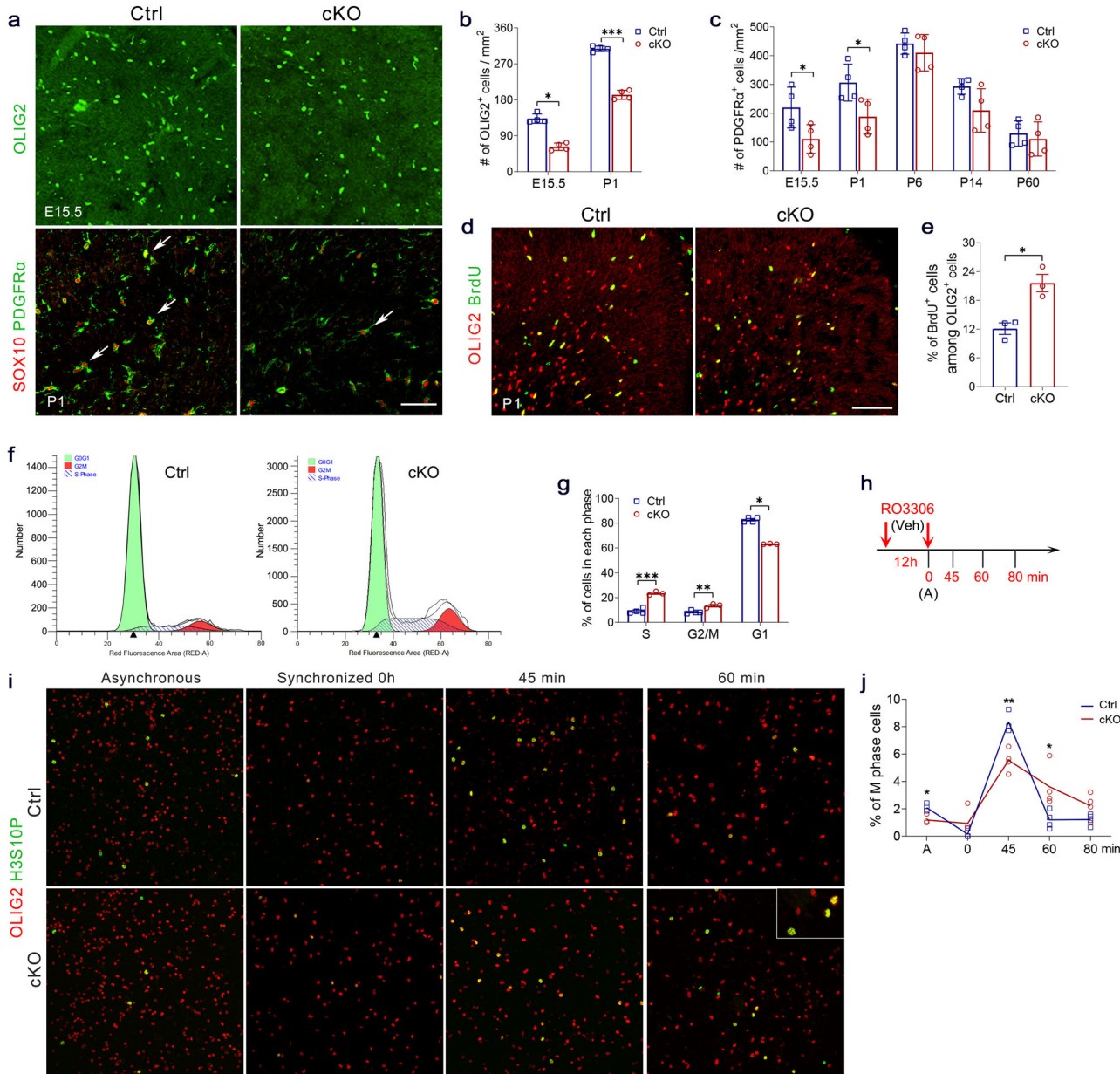

**Fig. 3 Proliferation and cell-cycle progression are defective in _Tet1_ cKO OPCs. a** Representative images of OLIG2 staining for embryonic day 15.5 (E15.5) and SOX10/PDGFRα staining for postnatal day 1 (P1) control and _Tet1_ cKO cortex. Scale bar, 100 μm. **b** Quantification of OLIG2+ cells in cortex from E15.5 and P1 control and _Tet1_ cKO mice. Data were Means ± SEM (_n_ = 4 animals per group). E15.5: *, Mann–Whitney _U_-test, _z_ = −2.309, _p_ = 0.021 compared to control; P1: ***, Two-tailed unpaired _t_-test, _t_ = 17.254, _df_ = 6, _p_ < 0.0001 compared to control. **c** Quantification of PDGFRα+ cells in cortex from E15.5, P1, P6, P14, and P60 control and _Tet1_ cKO mice. Data were Means ± SEM (_n_ = 4 animals per group). Two-tailed unpaired _t_-test, E15.5: *, _t_ = 2.519, _df_ = 6, _p_ = 0.045; P1: *, _t_ = 2.682, _df_ = 6, _p_ = 0.036; P60: _t_ = 0.522, _df_ = 6, _p_ = 0.620. Two-tailed unpaired separate variance estimation _t_-test, P6: _t_ = 0.871, _df_ = 4.823, _p_ = 0.425; P14: _t_ = 2.085, _df_ = 3.770, _p_ = 0.110 compared to control. **d** Representative images of P1 control and _Tet1_ cKO cortex immunostained for BrdU and OLIG2. Scale bar, 100 μm. **e** Quantification of BrdU+ cells within OLIG2+ OPC population in control and _Tet1_ cKO brains. Data were Means ± SEM (_n_ = 3 animals per group). *, Two-tailed unpaired _t_-test, _t_ = −4.380, _df_ = 4, _p_ = 0.012 compared to control. **f** Representative flow cytometry traces of propidium iodide-stained OPCs from control and _Tet1_ cKO mice. **g** Quantification of the percentage of OPCs from control and _Tet1_ cKO mice in each phase of the cell cycle. Data were Means ± SEM (_n_ = 4 independent cultures for the control group and _n_ = 3 independent cultures for _Tet1_ cKO group). Two-tailed unpaired _t_-test, S: ***, _t_ = −10.556, _df_ = 5, _p_ = 0.00013; G2/M: **, _t_ = −4.506, _df_ = 5, _p_ = 0.006; Mann–Whitney _U_-test, G1: *, _z_ = −2.141, _p_ = 0.032 compared to control.
**h** Diagram showing cell cycle synchronization assay. RO3306, a CDK1 inhibitor, arrests the cell cycle at the G2 phase. Cultures were fixed for immunostaining at A point (Vehicle treatment for 12 h), 0 (RO3306 treatment for 12 h), and 45-, 60-, 80-min after RO3306 withdrawal. **i** Representative H3S10P and OLIG2 double immunostaining for asynchronous and synchronized OPCs from control and _Tet1_ mutant. Cells were arrested at the G2/M phase with RO3306, 0 h, and released with fresh medium for 45- and 60-min. Enlarged inserts indicate H3S10P+ metaphase nuclei. Scale bar, 100 μm. **j** Quantification of the percentage of H3S10P+ metaphase cells among OLIG2+ cells in asynchronous (A), arrested (0 h), and released control and _Tet1_-mutant OPCs. Data were Means ± SEM (_n_ = 3 independent experiments for control group at 45 min, _n_ = 4 independent experiments for control at other time points and for _Tet1_ cKO group). Mann–Whitney _U_-test, A: *, _z_ = −2.337, _p_ = 0.019; 0 min: ns, _z_ = −1.692, _df_ = 6, _p_ = 0.091; Two-tailed unpaired _t_-test, 45 min: **, _t_ = 4.469, _df_ = 5, _p_ = 0.0066; 60 min: *, _t_ = 2.897, _df_ = 6, _p_ = 0.0275; 80 min: ns, _t_ = 1.971, _df_ = 6, _p_ = 0.0962.

phosphorylated histone H3 serine 10 (H3S10P), was used for immunostaining. In control OPCs, the percentage of H3S10P$^+$ cells among OLIG2$^+$ cells peaked at 45 min post-release, indicating that most cells had progressed into the early stages of mitosis (Fig. 3i, j). In contrast, less *Tet1* cKO OPCs were positive for H3S10P at this time point. At 60 min post-release, control cells had progressed into anaphase as most H3S10P staining disappeared in OLIG2$^+$ cells. In the case of *Tet1* cKO OPCs, we find that there is a delay in M-phase entry and withdrawal after cell cycle release (Fig. 3i, j). Thus, these results suggest that *Tet1* deletion results in a slower cell-cycle progression in OPCs, specifically during the G2-M transition, which likely leads to the observed reduction of OPC numbers in *Tet1* cKO animals.

In addition, we noted that *Tet1* deletion in *Tet1* cKO mice did not substantially alter the number of other neural cell types in the brain. Western blot and immunostaining with DCX, a marker for newly generated neurons; NeuN, a mature neuron marker; and ALDH1L1, an astrocyte marker, revealed comparable neuron and astrocyte pools between controls and *Tet1* mutants (Supplementary Fig. 7c–g). Meanwhile, we validated that TET1 expression in astrocytes and neurons was not significantly decreased in *Tet1* cKO mice (Supplementary Fig. 7h, i). Taken together, our data suggest that the abnormal cell-cycle progression of OPCs and delayed OPC differentiation contribute to the reduced OL numbers and hypomyelination in juvenile *Tet1*-deficient mice.

**TET1 regulates the transition from OPCs to OLs.** To determine if defects in OL differentiation caused by TET1 deletion are cell-autonomous, we isolated primary OPCs from the neonatal cortices of control and *Tet1* cKO pups and then treated them with T3 to induce differentiation. Immunostaining results showed that the percentage of CNP$^+$ and MBP$^+$ cells among OLIG2$^+$ cells in *Tet1*-deficient OPCs were significantly decreased when compared with control OPCs at 3 DIV (day in vitro) and 5 DIV (Fig. 4a–c), suggesting that *Tet1*-depleted OPCs are intrinsically reduced in their differentiation capacity.

Since the impaired myelination in *Tet1* cKO mice may result from the downsized OPC pool, to further confirm the role of TET1 in OL differentiation after birth, we bred *Tet1*$^{flox/flox}$ mice with *NG2-CreER* line[35], an OPC-specific tamoxifen-inducible Cre line to generate *Tet1*$^{flox/flox}$; *NG2-CreER* (*Tet1* OPC-iKO) animals. The *Tet1* OPC-iKO mice were treated daily with tamoxifen from P2 through P5 to induce *Tet1* deletion (Fig. 4d) and double immunostaining confirmed TET1 loss in SOX10$^+$ cells (Fig. 4e, f). Heterozygous littermates (*Tet1*$^{flox/+}$; *NG2-CreER*) were served as controls. In the P7 corpus callosum, more PDGFRα$^+$/Ki67$^+$ cells and less MBP$^+$ cells were observed in *Tet1* OPC-iKO mice (Fig. 4g–j). Moreover, a reduction in CC1$^+$ OLs and MBP intensity was observed in corpus callosum from P14 *Tet1* OPC-iKO mice (Fig. 4k, l). These results indicate that TET1 is required for the transition from OPCs to OLs.

In *Tet1* cKO mice, wherein *Tet1* is deleted in the *Olig1-Cre* expressing early precursors such as pri-OPC from the embryonic stages, we observed a decrease in the number of OPCs, suggesting that TET1 regulates the transition of OPCs from the pri-OPC state. However, in *Tet1*-iKO mice, wherein *NG2-CreER* is expressed in the committed OPCs, we observed a defect in OPC differentiation and a corresponding increase in undifferentiated OPCs, suggesting that TET1 deletion impairs OPC differentiation into mature OLs and maintains them in the OPC state. Thus, TET1 is involved in the early specification and later differentiation, producing stage-specific effects on OPC numbers.

We further performed the PPI test in *Tet1*-iKO mice at P42, which were treated daily with tamoxifen from P2 through P7 to

induce *Tet1* deletion in NG2$^+$ OPCs. Similar to *Tet1* cKO mice, the startle response was normal in *Tet1*-iKO mice, but the percentage of PPI was significantly attenuated (Supplementary Fig. 8a, b). These observations suggest that *Tet1* ablation in OL lineage cells leads to behavioral deficiency with impaired sensorimotor gating ability.

To further assess if TET1 deletion alters OPC fate or differentiation in adult mice, we used R26-EYFP reporter mice to test OL lineage progression in *Tet1*-iKO mice (*Tet1*$^{flox/flox}$; *NG2-CreER*; R26-EYFP). We performed tamoxifen-mediated deletion at P30 in *NG2-CreER Tet1*-iKO; R26-EYFP mice and analyzed the GFP reporter positive *Tet1*-iKO cells at P60. We did not detect any significant alteration in the number of OPCs (PDGFPα$^+$) and mature OLs (CC1$^+$) in *Tet1*-iKO mice (Supplementary Fig. 8c–f). Thus, these data suggest that *Tet1* ablation did not change OPC fate or differentiation in adult mice.

**Efficient remyelination requires TET1 function.** Given the critical role of TET1 in early OL development, we reasoned that TET1 should also be required in the adult brain for remyelination after an injury that results in demyelination. We induced demyelinated lesions in the corpus callosum via stereotaxic guided lysolecithin (LPC) injections (Fig. 5a). LPC induces rapid myelin breakdown followed by myelin regeneration through an OPC recruitment phase at 7 days post-lesion (7 dpl) induction and a remyelination phase at 14 dpl. To evaluate the role of TET1 in remyelination, we used *Tet1*$^{flox/flox}$; *NG2-CreER* (*Tet1* OPC-iKO) animals mentioned above. To induce recombination in adult mice, 8-week-old *Tet1* OPC-iKO mice were injected daily with tamoxifen for 8 days, starting 3 days prior to LPC injection in the corpus callosum (Fig. 5b). Brains were harvested at 7, 14, and 21 dpl from *Tet1* OPC-iKO mice and controls. Compared to control mice injected with vehicle, TET1$^+$ cell numbers were increased substantially in the lesion site at 7 dpl (Fig. 5c, d). In particular, the expression levels of TET1 in OLIG2$^+$ cells were higher after LPC treatment (Fig. 5c, d). In *Tet1* OPC-iKO mice, immunostaining showed ~97% OLIG2$^+$ cells lack TET1 expression at 7 dpl, indicating the efficient deletion of *Tet1* by *NG2-Cre* (Fig. 5c–e). Notably, TET1$^+$ OLIG2$^+$ OL lineage cells increased in the lesion at 7 dpl, while TET1 expression was detected in a subset of IBA1$^+$ microglia and NESTIN$^+$ cells, but not in GFAP$^+$ astrocytes (Supplementary Fig. 8g).

To determine the extent of remyelination, we examined the expression of OPC markers and myelin-related genes. Loss of *Tet1* did not appear to impair the recruitment of PDGFRα$^+$ OPCs, and the numbers of OPCs in the lesions were comparable between control and *Tet1* OPC-iKO mice during the recruitment phase at 7 dpl (Fig. 5 f, g). In contrast, there were significantly fewer GST-pi$^+$ differentiating OLs in the lesion site during the remyelination phase at 14 and 21 dpl in *Tet1* OPC-iKO mice relative to controls (Fig. 5 f, h). Consistent with a reduction in the number of differentiating OLs, MBP was also reduced in *Tet1*-iKO lesions compared to those of controls at 21 dpl (Fig. 5i). Notably, far fewer remyelinated axons were detected in the lesions of *Tet1*-iKO mice than controls at 28 dpl examined by EM (Fig. 5j, k). Furthermore, the thicknesses of newly generated myelin sheaths around axons were significantly reduced in *Tet1*-iKO mutants (Fig. 5l). These observations indicate that TET1 is required for the proper remyelination in the context of white matter injury.

**Transcriptome alterations and a genome-wide decrease of 5hmC in *Tet1*-ablated OPCs.** To investigate the molecular underpinnings of the observed defects in early OL development, we compared the RNA transcriptomes of OPCs cultured from

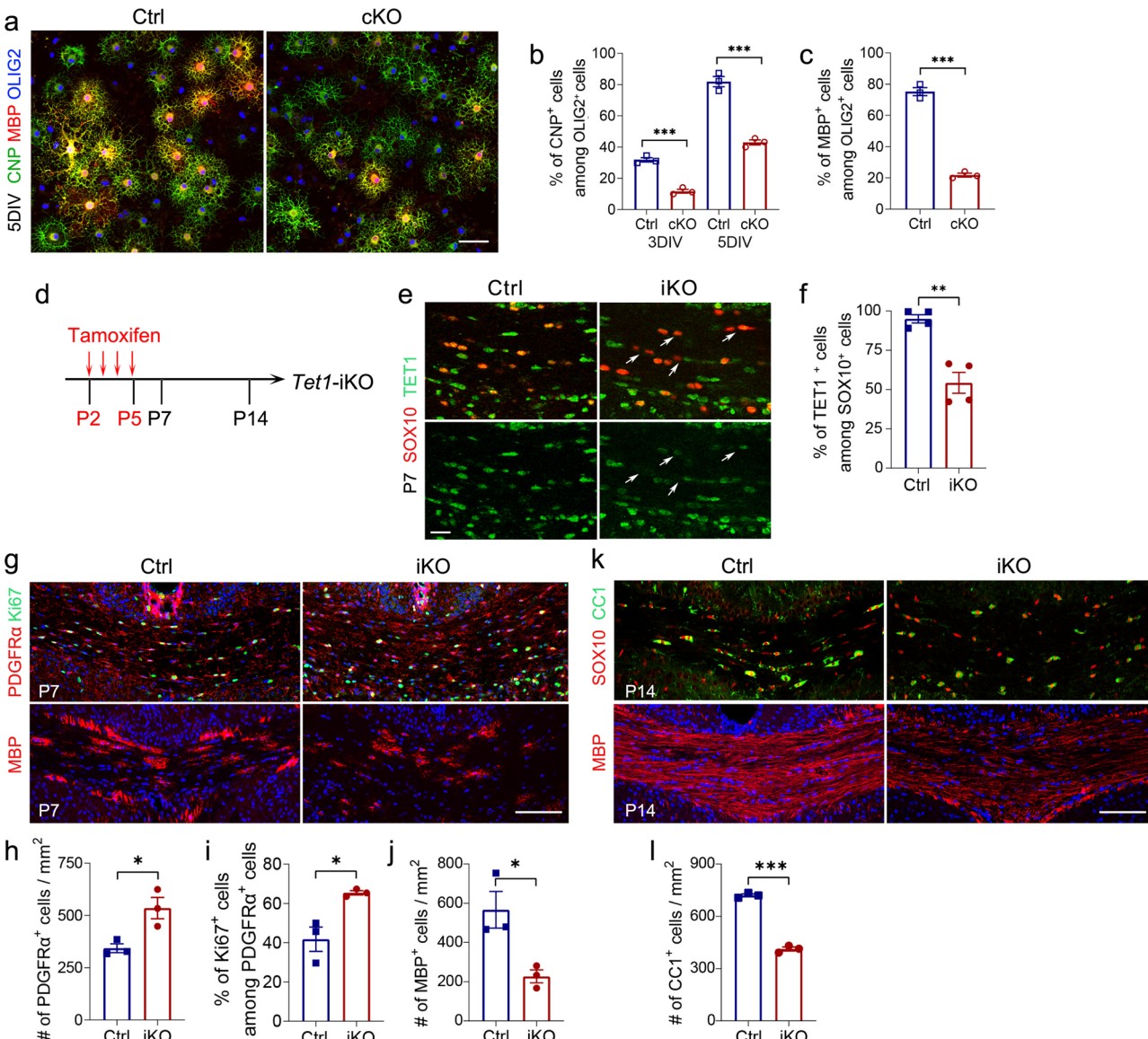

**Fig. 4 OPC differentiation is impaired in *Tet1* OPC-iKO mice. a** Representative images of OL cultures from control and *Tet1* cKO mice immunostained with CNPase, MBP, and OLIG2 at 5 days in vitro (5 DIV). Scale bar, 50 μm. **b** Quantification of CNPase⁺ cells among OLIG2⁺ cells in the cultures from control or *Tet1* cKO mice at 3 or 5 DIV. Data were Means ± SEM (*n* = 3 of independent cultures in each group). Two-tailed unpaired *t*-test, 3 DIV: ***, *t* = 10.716, *df* = 4, *p* = 0.00043; 5 DIV: ***, *t* = 10.603, *df* = 4, *p* = 0.000448 compared to control. **c** Quantification of MBP⁺ cells among OLIG2⁺ cells in the cultures from control or *Tet1* cKO mice at 5 DIV. Data were Means ± SEM (*n* = 3 of independent cultures in each group). Two-tailed unpaired *t*-test, ***, *t* = 18.644, *df* = 4, *p* = 0.000049 compared to control. **d** Diagram showing Tamoxifen administration to induce the Cre recombination in *Tet1* OPC-iKO (*Tet1* flox/flox; *NG2-CreER*) mice. **e** Representative images of P7 control and *Tet1*-iKO mice corpus callosum stained for TET1 and SOX10. Arrows indicate SOX10⁺ cells that show reduced levels of TET1 in *Tet1*-iKO image. Scale bar, 20 μm. **f** Percentages of TET1⁺ cells among SOX10⁺ oligodendrocytes in P7 corpus callosum. Data were Means ± SEM (*n* = 4 animals per group). Two-tailed unpaired separate variance estimation *t*-test, **, *t* = 5.707, *df* = 3.933, *p* = 0.005 compared to control. **g** Representative images of control and *Tet1* OPC-iKO P7 corpus callosum stained for PDGFRα/Ki67 or MBP. Scale bar, 100 μm. **h–j** Quantification of the density of PDGFRα⁺ cells (**h**), the percentage of Ki67⁺ cells among PDGFRα⁺ cells (**i**), and the density of MBP⁺ cells at P7 (**j**). Data were presented as Means ± SEM (*n* = 3 animals per group). Two-tailed unpaired *t*-test, **h**: *, *t* = −3.489, *df* = 4, *p* = 0.025 compared to control. **i**: *, *t* = −3.757, *df* = 4, *p* = 0.020 compared to control. **j**: *, *t* = 3.426, *df* = 4, *p* = 0.027 compared to control. **k** Representative images of control and *Tet1* OPC-iKO P14 corpus callosum stained for CC1/SOX10 and MBP. Scale bar, 100 μm. **l** Quantification of the density of CC1⁺ cells at P14. Data were presented as Means ± SEM (*n* = 3 animals per group). Two-tailed unpaired *t*-test, ***, *t* = 22.136, *df* = 4, *p* = 0.000025 compared to control.

control and *Tet1* cKO neonates. There were ~1880 genes down-regulated and 881 genes upregulated in *Tet1* mutants compared with controls (FDR < 0.05, Log₂ > 1 or < −1) (Supplementary Table 2), suggesting that the predominant effect of *Tet1* was transcriptional activation. As indicated by GSEA, among the top downregulated TET1 targets were those pertinent to cell division, extracellular matrix, and OL differentiation (Fig. 6a–d), which are

in consistent with the abnormal cell cycle and deficient myelin formation in *Tet1* mutants. In contrast, GO terms associated with the mitochondria gene module and immune-related functions were upregulated in *Tet1* cKO OPCs (Fig. 6a, b, e, f). The observation that TET1-activated genes are related to OL differentiation may reflect a cell-type-specific transcriptional regulation for TET1 in OL differentiation. Quantitative PCR confirmed

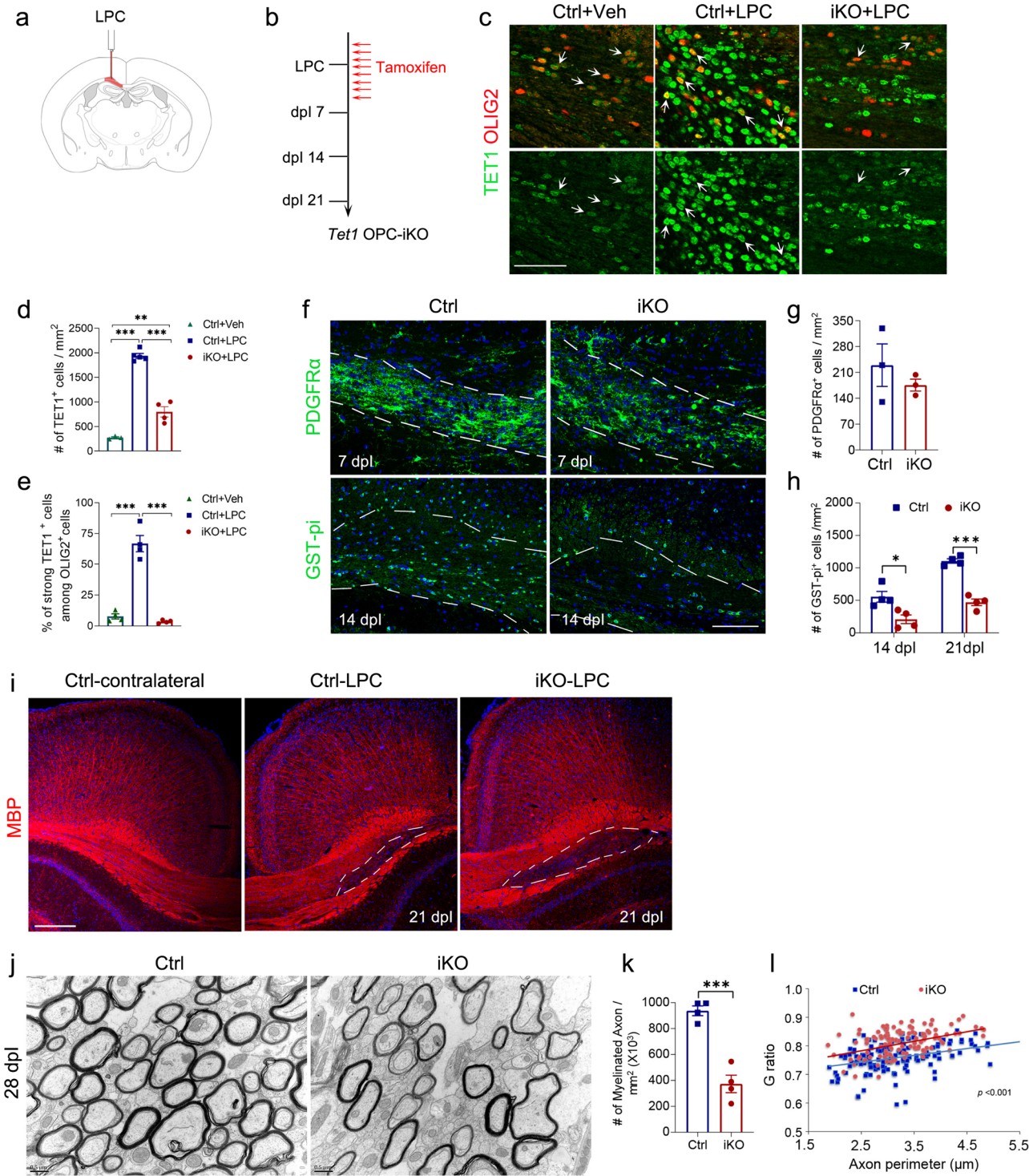

representative gene expression changes associated with these gene set enrichments in the RNA-seq data (Fig. 6g–i). Specifically, OL differentiation and myelination-associated genes *Mbp*, *Plp1*, *Myrf*, *Enpp2*, *Cnp*, *Cldn11*, *Ugt8a*, *Kif11*, and *Bcas1* were markedly downregulated in *Tet1*-deficient OPCs (Fig. 6g) as were genes involved in cell-cycle regulation, e.g., *Ccnb1*, *Cdc25b*, and *Cdc25c* (Fig. 6h and Supplementary Fig. 6j, k). The transcriptome landscape alterations in *Tet1* cKO OPCs were in line with the observations that *Tet1* depletion led to cell-cycle progression defects and hypomyelination phenotypes.

Since TET1 mediates DNA hydroxymethylation/demethylation, we next tested the level of 5hmC in OLs from *Tet1* mutants.

In P27 brain sections, immunostaining of 5hmC simultaneously with the OL marker CC1 revealed a striking reduction in 5hmC intensity in *Tet1* cKO OLs (Fig. 7a, b), which strongly suggested that 5hmC is involved in TET1-mediated regulation of OL differentiation. To further compare the genome-wide 5hmC distributions, we performed hMeDIP-seq in OPC cultures from controls and *Tet1* mutants. *Tet1* cKO OPCs showed a dramatic reduction in 5hmC peak signals compared to controls (Supplementary Fig. 9a). In both groups, most 5hmC peaks resided in intergenic regions; less than 40% of peaks were within gene bodies of annotated RefSeq genes (Supplementary Fig. 9b). This is different from the distribution pattern in mouse embryonic stem

**Fig. 5 Impaired CNS remyelination in *Tet1* OPC-iKO mice. a** Diagram of the brain showing the lysolecithin (LPC) injection site. **b** Schedule for Tamoxifen administration and LPC injection in *Tet1* flox/flox; *NG2CreER* mice (*Tet1* OPC-iKO). **c** Representative images of TET1 immunostaining in the corpus callosum of non-lesion vehicle injection region of control mice (Ctrl + Veh), LPC-induced lesion region of control mice (Ctrl + LPC), and *Tet1*-iKO mice (iKO + LPC) at 7 days post-lesion (7 dpl). Arrows indicate OLIG2+ cells. Note the upregulated TET1 expression in Ctrl + LPC mice, but the decreased TET1 expression in iKO + LPC mice, indicating the effective ablation of TET1 in *Tet1*-iKO mice. Scale bar, 50 μm. **d** Quantification of TET1+ cell density in LPC lesion sites at 7 dpl. Data were presented as Means ± SEM ($n = 3$ animals for Ctrl + Veh group, $n = 5$ animals for Ctrl + LPC group, $n = 4$ animals for iKO + LPC group). One-way ANOVA, $F_{(2,9)} = 146.116$, $p < 0.00001$. Tukey's multiple comparisons test: ***, Ctrl + Veh vs. Ctrl + LPC: $q = 22.68$, $df = 9$, $p < 0.0001$, **, Ctrl + Veh vs. iKO + LPC: $q = 6.842$, $df = 9$, $p = 0.0024$, ***, Ctrl + LPC vs. iKO + LPC: $q = 16.90$, $df = 9$, $p < 0.0001$. **e** Quantification of the percentages of strong TET1+ among OLIG2+ cells in LPC lesion sites at 7 dpl. Data were presented as Means ± SEM ($n = 4$ animals for each group). One-way ANOVA, $F_{(2,9)} = 69.852$, $p < 0.000003$. Tukey's multiple comparisons test: ***, Ctrl + Veh vs. Ctrl + LPC: $q = 14.57$, $df = 9$, $p < 0.0001$, ns, Ctrl + Veh vs. iKO + LPC: $q = 0.1852$, $df = 9$, $p = 0.9906$, Ctrl + LPC vs. iKO + LPC: ***, $q = 14.38$, $df = 9$, $p < 0.0001$. **f** Representative images of lesion regions from control and *Tet1*-iKO mutants at 7 dpl and 14 dpl. Samples were immunolabeled for PDGFRα and GST-pi, respectively. Dashed line indicates the border of the lesion site. Scale bars, 100 μm. **g** Quantification of PDGFRα+ OPCs in LPC lesion sites at 7 dpl. Data were Means ± SEM ($n = 3$ animals in each group). Two-tailed unpaired *t*-test, $t = 0.905$, $df = 4$, $p = 0.416$ compared to control. **h** Quantification of GST-pi+ OLs in LPC lesion sites at 14 and 21 dpl. Data were Means ± SEM ($n = 4$ animals in each group). Two-tailed unpaired *t*-test, 14 dpl: *, $t = 3.241$, $df = 6$, $p = 0.018$; 21 dpl: ***, $t = 10.281$, $df = 6$, $p = 0.000049$ compared to control. **i** Contralateral side and LPC lesion regions from control and *Tet1*-iKO corpus callosum at 21 dpl. Samples were immunostained for MBP. Scale bar, 200 μm. **j** Representative images of electron micrographs of lesion regions from control and *Tet1*-iKO mutants at 28 dpl. Scar bar, 0.5 μm. **k** The number of myelinated axons in lesion regions from control and *Tet1*-iKO mutants at 28 dpl. Data were Means ± SEM ($n = 4$ slides from three animals per group). Two-tailed unpaired *t*-test, ***, $t = 7.294$, $df = 6$, $p = 0.00034$ compared to control. **l** G ratios vs. axonal perimeters in lesion regions from control and *Tet1*-iKO mutants at 28 dpl. Friedman M test, $\chi^2 = 47.641$, $df = 1$, $p < 0.0001$. $p_{between\ group} < 0.001$ (>130 myelinating axon counts from three animals each genotype).

cells[18] and neurons[36]. After plotting the distribution of 5hmC peaks over RefSeq genes, we found that 5hmC was reduced near the TSS and transcription terminal site (TTS) in control OPCs and that *Tet1* depletion caused reductions of 5hmC, especially in intragenic regions, promoter regions, and TTS regions (Fig. 7c and Supplementary Fig. 9b). Heatmap clustering of the 5hmC peak distributions 5-kb upstream and downstream of TSSs revealed five groups; levels of 5hmC signals were lower in all five groups in *Tet1* cKO samples than in the control samples (Fig. 7d). Further analysis showed that most differentially hydroxymethylated regions also had a low CpG density of less than 1 CpG per 100 bp (Supplementary Fig. 9c).

We next examined the correlation between hydroxymethylation and gene expression. By integrating RNA-seq data with hMeDIP-seq data, we observed that among the genes that had lower hydroxymethylation in gene body regions in the *Tet1* cKO mice than in controls, 12.83% was downregulated (1026 of 7998 genes) and 4.89% was upregulated (391 of 7998 genes) (Supplementary Fig. 9d). The percentages were comparable in promoter regions that were differentially hydroxymethylated with 13.46% of genes downregulated and 6.42% upregulated genes (Supplementary Fig. 9e). These observations indicate that DNA hydroxymethylation is positively correlated with gene expression in OLs.

Among the genes that showed both downregulated mRNA expression and decreased gene body 5hmC levels in *Tet1* cKO mice, GO analysis revealed that a majority of them are associated with OL differentiation, cell proliferation, and extracellular matrix (Fig. 7e, f). For example, *Mbp*, *Mobp*, and *Cnp* are myelin-related genes; *Tcf7L2*, *Myrf*, and *Enpp2/6* are involved in the regulation of OL differentiation in a stage-specific manner within different transcriptional circuitries[37,38]. Snapshots of 5hmC profiles of representative myelination genes and cell division genes showed reduced 5hmC levels in *Tet1* cKO group (Fig. 7g and Supplementary Fig. 9f, g). To further test the genome-wide 5hmC landscape changes during OPC-OL transition, we compared hMeDIP-seq data in OLs with that in OPCs. Interestingly, we find increased 5hmC intensity in OL compared to OPC (Supplementary Fig. 10), which indicates that the changes in 5hmC epigenetic marks might contribute to OPC-OL transition. Together, these results are highly in consistent with the hypomyelination phenotypes in *Tet1* cKO

mice and indicate the significance of TET1-5hmC mechanisms in OL homeostasis.

To further identify potential TET1 direct targets, we performed Cut & Run assays[39] for high-resolution mapping of TET1 DNA binding sites. Most TET1 binding sites overlapped with 5hmC modified regions (Fig. 8a), suggesting that TET1 directly binds to these target genes and facilitates hydroxymethylation in the regulatory elements. GO analysis for TET1 and 5hmC overlapped targets in OPCs revealed genes associated with OL differentiation, calcium ion transport, and cell proliferation (Fig. 8b). Furthermore, TET1 binding profiles with TET1 antibody in representative genes were in accordance with 5hmC peaks, and associated with the activating histone mark H3K27ac (Fig. 8c), indicating a transcriptional activation role of TET1 in addition to its DNA methylcytosine dioxygenase activity in the regulation of gene transcription.

**Impaired calcium transport leads to differentiation defects in OPC cultures from *Tet1* cKO mice.** When searching for TET1-5hmC regulated factors that may involve in OL development and homeostasis, we noticed that there was a cluster of calcium transporter genes among the downregulated and hypo-hydroxymethylated genes in *Tet1* cKO group (Figs. 6a, b and 7e, f, h, i). The decreased expression of these genes in *Tet1* cKO mice was confirmed by qRT-PCR assays (Fig. 7j). CACNA1a, CACNA1c, CACNA2d1, CACNB4, and CACNG5 are plasma membrane voltage-operated $Ca^{2+}$ channels (VOCCs) that are expressed in OPCs and contribute to calcium dynamics in these cells[22,40]. In particular, calcium influx mediated by CACNA1c, also known as $Ca_v1.2$, is required for OL differentiation[41,42]. Another target gene *Itpr2*, which encodes a type 2 IP3 receptor, localized to the ER and expressed exclusively in postmitotic OLs[43,44], also showed decreased mRNA expression in OPCs from *Tet1* cKO mice compared to controls (Fig. 7j). These data indicate that TET1-5hmC targets calcium transport genes and may regulate calcium dynamics in OLs. Similar to a recent observation[23], we also observed several members of the solute carrier (*Slc*) gene family as TET1-5hmC targets, e.g., *Slc24a2*, *Slc8a1*, *Slc12a2*, and *Slc7a14* (Fig. 7h–j and Supplementary Fig. 9g). This result indicates that TET1-5hmC may control OL function through multiple targets.

To evaluate the intracellular $Ca^{2+}$ concentration ($[Ca^{2+}]_i$) fluctuation in OPC culture, a Fluo4-based $Ca^{2+}$-imaging approach

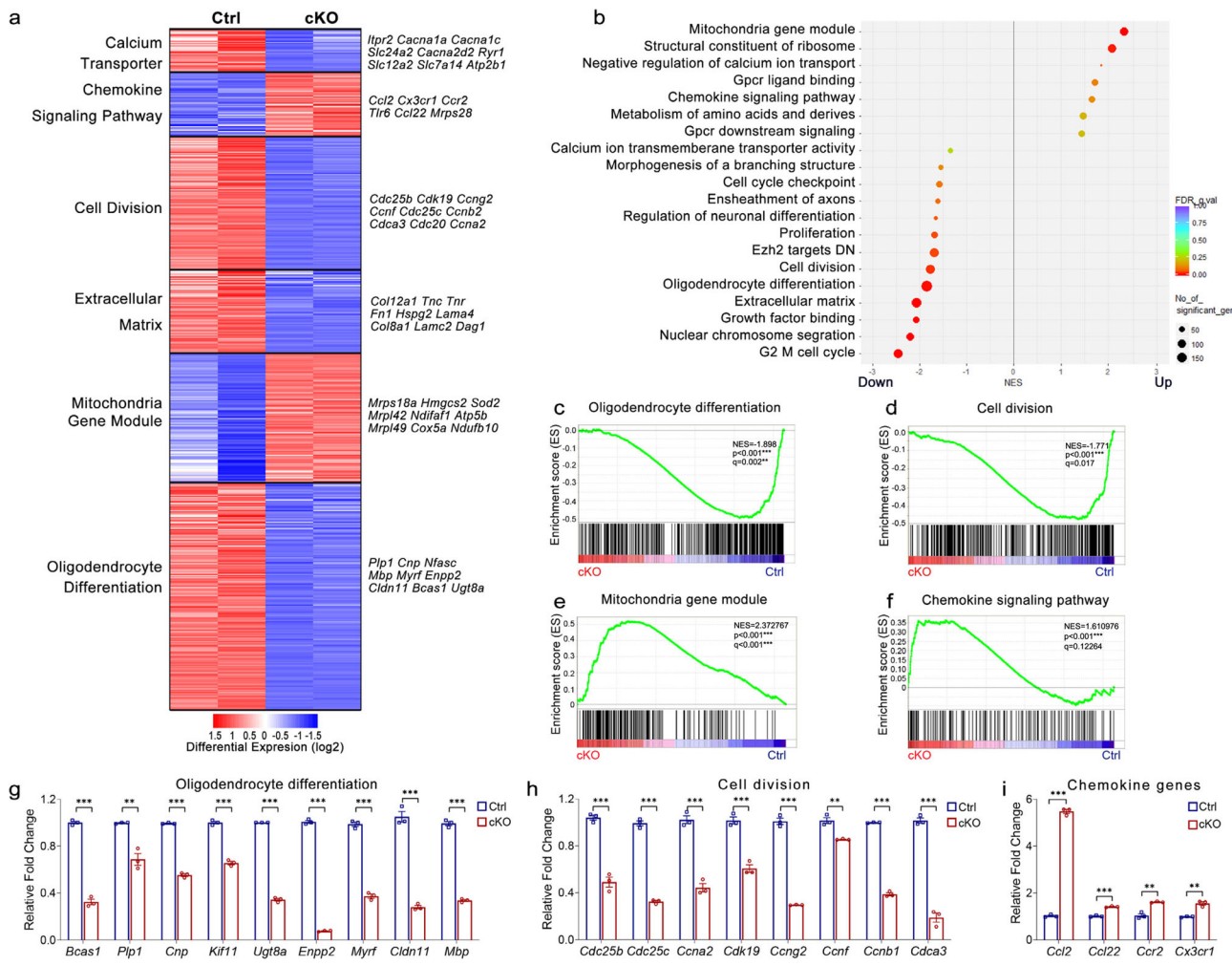

**Fig. 6 Lack of *Tet1* alters the transcriptome profiles in OPCs. a** Heatmap of RNA-seq data from purified OPCs shows categories of differentially expressed genes between control and *Tet1* cKO mice. Each genotype was repeated once. **b** GSEA analysis of the most differentially regulated genes in OPC cultures from *Tet1* mutant and control mice. **c–f** GSEA plots of genes involved in oligodendrocyte differentiation (**c**), cell division (**d**), mitochondria genes (**e**), and chemokine signaling pathway genes (**f**) in control and *Tet1*-mutant OPCs. **g–i** Quantitative real-time PCR of oligodendrocyte differentiation genes (**g**), genes involved in cell division and G2/M cell cycle regulation (**h**), and chemokine genes (**i**) in control and *Tet1*-mutant OPCs. Data were Means ± SEM of transcript levels relative to control after normalization from $n = 3$ independent experiments each performed in triplicate. **g** Two-tailed unpaired *t*-test. *Bcas1*: ***, $t = 25.375$, $df = 4$, $p = 0.000014$; *Plp*: **, $t = 6.214$, $df = 4$, $p = 0.0034$; *Cnp*: ***, $t = 35.28$, $df = 4$, $p < 0.0001$; *Kif11*: ***, $t = 18.50$, $df = 4$, $p = 0.00005$; *Ugt8a*: ***, $t = 56.64$, $df = 4$, $p < 0.0001$; *Enpp2*: ***, $t = 74.57$, $df = 4$, $p < 0.0001$; *Myrf*: ***, $t = 23.955$, $df = 4$, $p = 0.000013$; *Cldn11*: ***, $t = 16.173$, $df = 4$, $p = 0.000086$; *Mbp*: ***, $t = 33.299$, $df = 4$, $p = 0.000005$. **h** Two-tailed unpaired *t*-test. *Cdc25b*: ***, $t = 11.384$, $df = 4$, $p = 0.000344$; *Cdc25c*: ***, $t = 25.527$, $df = 4$, $p = 0.000014$; *Ccna2*: ***, $t = 12.123$, $df = 4$, $p = 0.000266$; *Cdk19*: ***, $t = 9.264$, $df = 4$, $p = 0.0008$; *Ccng2*: ***, $t = 26.16$, $df = 4$, $p < 0.0001$; *Ccnf*: **, $t = 6.263$, $df = 4$, $p = 0.0033$; *Ccnb1*: ***, $t = 47.51$, $df = 4$, $p < 0.0001$; *Cdca3*: ***, $t = 17.992$, $df = 4$, $p = 0.000056$. **i** Two-tailed unpaired *t*-test. *Ccl2*: ***, $t = 47.22$, $df = 4$, $p < 0.0001$; *Ccl22*: ***, $t = -14.905$, $df = 4$, $p = 0.000118$; *Ccr2*: **, $t = -7.249$, $df = 4$, $p = 0.00192$; *Cx3cr1*: **, $t = 7.773$, $df = 4$, $p = 0.0015$.

was employed. ATP can mobilize $Ca^{2+}$ from ER stores or trigger $Ca^{2+}$ influx across the plasma membrane[45,46]. To first determine the intracellular calcium release, we used calcium-free aCSF to load Fluo4. In OPC cultures, the application of ATP induced a transient increase of $[Ca^{2+}]_i$ in both control and *Tet1* cKO groups (Fig. 9a, b and Supplementary Movies 1 and 2). The amplitude of ATP-induced calcium rise was significantly higher in control OPCs than in *Tet1*-mutant OPCs (Fig. 9c). This result indicates a deficit of intracellular calcium mobilization in *Tet1*-depleted OPCs. Additionally, we compared the relative free $Ca^{2+}$ level of internal stores between control and *Tet1* cKO OPCs with mag-Fluo4-AM, a low $Ca^{2+}$ affinity fluorescent indicator for the ER, as previously reported[47]. Living OPCs from both groups showed similar relative mag-Fluo4-AM fluorescence in the ER, as measured by spectro-fluorometer (Supplementary Fig. 11a, b).

To further examine the extracellular calcium influx in *Tet1* cKO OPCs, we used Bay K 8644, an L-type calcium channel (L-VOCC) agonist[48], to induce synchronous calcium transient in OPCs, which can be blocked by the specific L-VOCC inhibitor Verapamil (Supplementary Fig. 11c). In the mutant OPCs, $[Ca^{2+}]_i$ increase was of lower amplitude and the kinetics of onset were slower (Supplementary Fig. 11d, e), suggesting the attenuated function of membrane calcium channel in *Tet1*-ablated OPCs.

To investigate the consequences of impaired calcium rise in *Tet1*-deficient OPCs, we examined cell differentiation after activating calcium signaling. Consistent with the results of high $K^+$ application[42], three consecutive pulses (5 min/each) daily with 10 μM calcium channel agonist Bay K 8644 significantly promoted differentiation of OPCs and partially rescued the differentiation

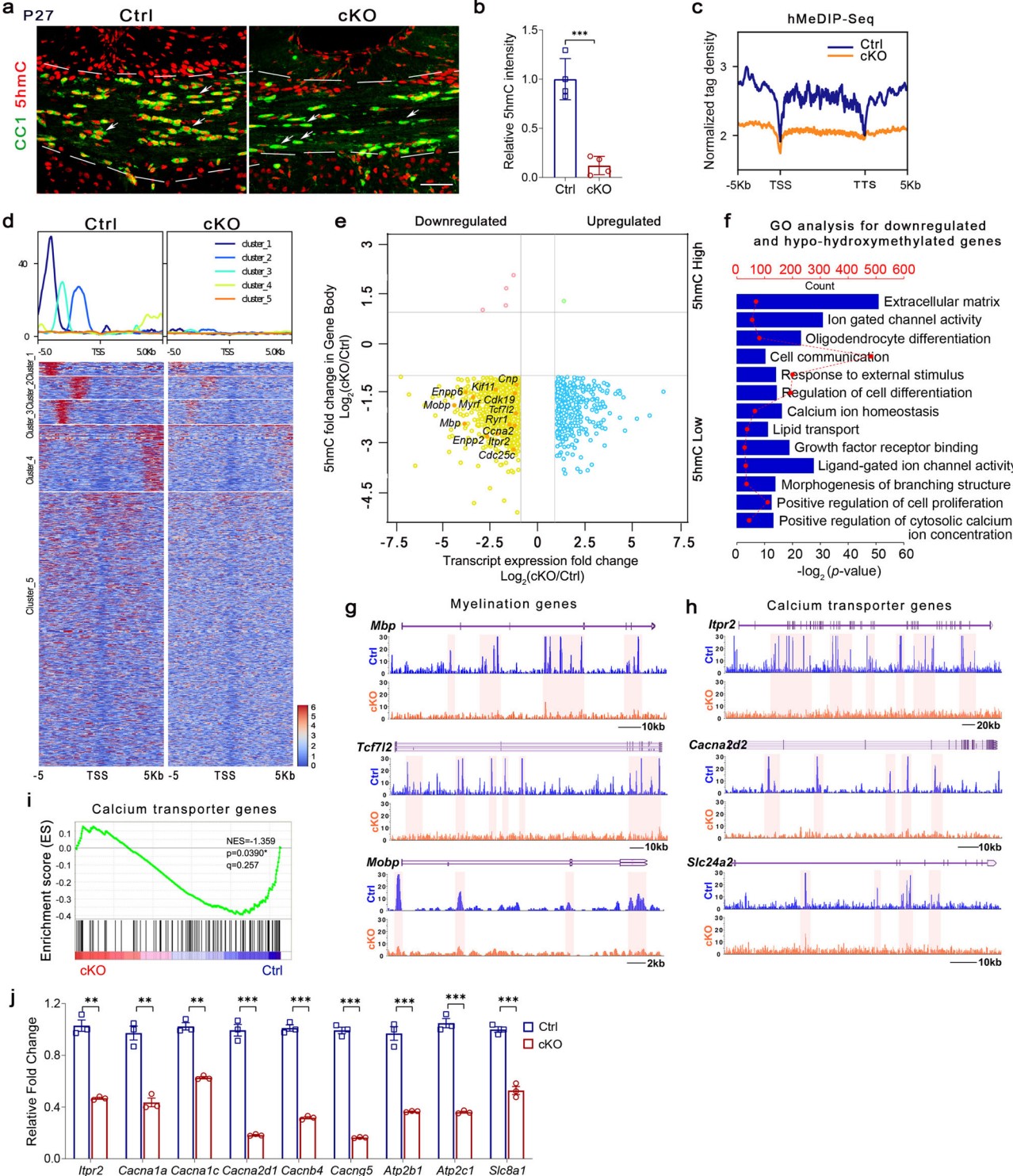

defects in *Tet1*-deficient OPCs as determined by qRT-PCR analysis of myelin genes and MBP$^+$ OL numbers (Supplementary Fig. 11f–h). These data indicate that TET1-modulated $[Ca^{2+}]_i$ rise play important roles in the OL differentiation process.

**Ablation ITPR2, a modulator for calcium release from ER, inhibits OL differentiation.** To further distinguish calcium signaling as TET1-5hmC target during OL differentiation, or the results of impaired differentiation in *Tet1* cKO mice, we then tested the function of ITPR2, one of TET1-5hmC target calcium transport genes, in myelination. As a modulator for calcium release

from ER, ITPR2 showed the highest expression in postmitotic newly formed OLs (OL-1DIV, PDGFRα$^-$/CNPase$^+$) (Fig. 9d and Supplementary Fig. 12a) as reported previously[22,43,44]. In vitro studies with RNAi methods revealed that transfection of siRNA against *Itpr2* in normal OL cultures significantly reduced the expression of myelin genes and impaired OL differentiation (Fig. 9e–g and Supplementary Fig. 12b, c). Additionally, we tested the effect of adenophostin-A (Ada), a high-affinity agonist of ITPR2[49], on rescuing impaired *Tet1* cKO OPC calcium rise and differentiation in cultures. Daily application of Ada for 2 days in *Tet1* cKO OPC cultures significantly increased the amplitude of

**Fig. 7 DNA hydroxymethylation of genes and transcript levels are correlated in oligodendrocytes from *Tet1* cKO mice. a** Representative images of 5hmC and CC1 immunostaining in the corpus callosum of P27 control and *Tet1* cKO mice. Attenuated level of 5hmC in CC1+ cells in *Tet1* mutant is indicated by arrows. Scale bar, 50 μm. **b** Quantification the fluorescence intensity of 5hmC staining in CC1+ cells from control and *Tet1* cKO brains. Data were Means ± SEM ($n = 4$ animals per group). Two-tailed unpaired *t*-test, ***, $t = 7.660$, $df = 6$, $p = 0.000259$ compared to control. **c** Normalized 5hmC tag density distribution from *Tet1* cKO and control OPCs. TSS transcription start site, TTS transcription terminal site. **d** Heatmap of 5hmC signal from ±5 kb of TSS at all annotated genes in control and *Tet1* cKO OPCs. **e** Quadrant plot of differentially hydroxymethylated peaks at gene bodies and differentially expressed genes (*Tet1* cKO vs. Control, $p < 0.05$). The x-axis is Log$_2$ fold change of transcript level, and the y axis is the difference in DNA hydroxymethylation. Dashed lines indicate a twofold difference. Genes indicated are involved in OL differentiation, cell division, and calcium transporter. **f** Gene ontology (GO) analysis for downregulated and hypo-hydroxymethylated gene groups in *Tet1*- ablated OPCs. **g**, **h** 5hmC enriched genes involved in myelination (**g**) and calcium transport (**h**) from control (blue) and *Tet1* cKO (orange) OPCs. Tracks were visualized with Mochiview v1.46. **i** GSEA plots of genes involved in calcium ion transmembrane transporter activity in control and *Tet1*-mutant OPCs. **j** Quantitative real-time PCR of genes involved in calcium ion transmembrane transporter activity in control and *Tet1* cKO OPCs. Data were Means ± SEM of transcript levels relative to control after normalization from $n = 3$ independent experiments each performed in triplicate. Two-tailed unpaired *t*-test. *Itpr2*: **, $t = 12.124$, $df = 4$, $p = 0.006$; *Cacna1a*: **, $t = 8.606$, $df = 4$, $p = 0.001002$; *Cacna1c*: **, $t = 13.36$, $df = 4$, $p = 0.0002$; *Cacna2d1*: ***, $t = 17.90$, $df = 4$, $p < 0.0001$; *Cacnb4*: ***, $t = 27.209$, $df = 4$, $p = 0.000011$; *Cacng5*: ***, $t = 32.76$, $df = 4$, $p < 0.0001$; *Atp2b1*: ***, $t = 11.47$, $df = 4$, $p = 0.0003$; *Atp2c1*: ***, $t = 18.896$, $df = 4$, $p = 0.000046$; *Slc8a1*: ***, $t = 12.47$, $df = 4$, $p = 0.0002$.

ATP-induced calcium rise (Fig. 9b, c). Similar to the above rescue assay with Bay K 8644 application, we revealed by immunostaining that the addition of Ada significantly increased the differentiation ability of *Tet1* cKO OPCs in culture (Supplementary Fig. 12d–f).

Given that TET1 regulates a set of genes in the calcium transport, activation of a single gene *Itpr2* might not be able to fully rescue the calcium defects in *Tet1* cKO mice. We then examined the in vivo function of ITPR2 on OL differentiation by generating *Itpr2* conditional knockout mice *Itpr2* cKO (*Itpr2*$^{flox/flox}$; *Olig1*-Cre$^{+/−}$). To assess Cre-mediated *Itpr2* depletion, we quantified ITPR2 expression in OLIG2+ cells from *Itpr2* cKO and control mice at P14. Immunostaining revealed that expression of ITPR2 in the corpus callosum was significantly lower in *Itpr2* cKO than in control mice (Fig. 9h, i). Quantitative real-time PCR further confirmed the specific ablation of *Itpr2* in purified OPCs, but not in astrocytes (Supplementary Fig. 12g). Examination of the expression of stage-specific OL lineage markers revealed that the percentage of CC1+ mature OLs was significantly reduced in *Itpr2* cKO cortices compared to controls at P7 and P14 (Fig. 9j, l), whereas the number of PDGFRα+ OPCs increased (Fig. 9j, k). Myelin protein expression was also substantially decreased in *Itpr2* cKO mice at P31 (Fig. 9m). EM of P16 optic nerves revealed that the number of myelinated axons was significantly reduced in *Itpr2* mutants (Fig. 9n, o), and the G ratio was higher in mutants (Fig. 9p). Similar to the phenotypes in adult *Tet1* cKO mice, the developmental deficiency of *Itpr2* mutants was recovered in adult animals, as indicated by the comparable number of CC1+ OLs and NG2+ OPCs between control and *Itpr2* mutant animals at P60 (Fig. 9k, l and Supplementary Fig. 12h). Together, these results suggest that as one of the target genes of TET1-5hmC, ITPR2 accumulates after cell cycle withdrawal and is involved in the initiation of OL differentiation, probably by effectively releasing calcium from ER.

## Discussion

**TET1 functions stage-specifically in OL development and remyelination.** Methylation of cytosine on CpG islands in the genome enables stable but reversible transcription repression and is critical for mammalian development[7,8]. Defects in the regulation of DNA methylation are associated with various neurological diseases[50,51]. TET enzymes catalyze the first step of DNA demethylation by oxidizing 5mC into 5hmC[10–12]. Strikingly, we detected a genome-wide change in the DNA demethylation landscape marked by 5hmC during OPC differentiation from NPCs, suggesting a role of TET-mediated DNA demethylation in the regulation of OL lineage progression.

We found that TET1, but not TET3, is critical for OPC proliferation, differentiation, and myelination during early animal development, suggesting a unique function of TET1 in oligodendrogenesis and subsequent myelinogenesis. Our observation is different from another study that *Tet1* depletion has little effect on OPC-to-OL differentiation in the young mouse brain (*Tet1*$^{loxp/loxp}$; *Olig1*Cre)[23]. The reasons for different phenotypes between two mouse lines are not immediately clear. We speculate that the use of different *Tet1*-floxed strains might contribute to the phenotypic discrepancy during early developmental stages. In our study, the *Tet1*-mutant mice lack exons 11–13, encoding the catalytic domain of TET1 enzyme[24], while *Tet1*-mutant mice lacking exon 4 (*Tet1* Δe4)[52] were used in the other study[23]. Distinct phenotypes and gene expression changes have been reported among individual *Tet1*-mutant mice lacking exon 4[52] or exon 11–12[24]. Thus, it is possible that the phenotype discrepancy observed in different *Tet1*-mutant mice might be due to the nature of the mutations disrupting specific *Tet1* isoforms, as suggested by the alternative splicing forms of the *Tet1* gene[53]. In addition, other possibilities such as differences in *Olig1*-Cre lines, mosaicism, and mouse backgrounds used in the studies might also contribute to the phenotypic discrepancy.

Despite early developmental defects, we noticed that the developmental myelin deficiencies recovered in adult *Tet1* cKO mice. TET1 expression is downregulated in OLs from old mice, and we observed weak TET1 expression in a population of OLs (~29%) in adult *Tet1* cKO mice, more than ~12% TET1+ OLs in *Tet1* mutants at P4. Thus, myelin deficiencies that recovered in adult *Tet1* cKO mice might be in part due to the escape from Cre-mediated *Tet1* depletion in a population of OL lineage cells. Similar observations have been suggested in other studies, e.g., for the recovered myelin deficiency in adult *Dicer1* cKO mice[54]. In addition, it is also possible that the myelination process is not affected in adult OLs lacking TET1 expression, therefore, TET1 may not be essential for myelinogenesis in mature OLs. Furthermore, the remyelination capacity after injury was compromised in adult *Tet1* OPC-iKO brains, suggesting that TET1 is also critical for the myelin regeneration process.

Although OL differentiation defects were not due to increased apoptosis in the *Tet1*-mutant brain, we found that OPC cell cycle progression, especially G2 to M phase transition, was impaired in developing *Tet1* cKO animals. In line with this result, we showed that factors involved in cell-cycle regulation, *Ccng2*, *Cdca3*, *Ccna2*, *Ccnb1*, and *Cdc25b*[55–59], were 5hmC enriched genes in OPCs and were downregulated in *Tet1* cKO OPCs. In particular, CCNB1 (Cyclin B1) is the regulatory subunit of CDK1 serine/threonine kinase, and accumulation of CCNB1 is a prerequisite for mitotic initiation in the late G2

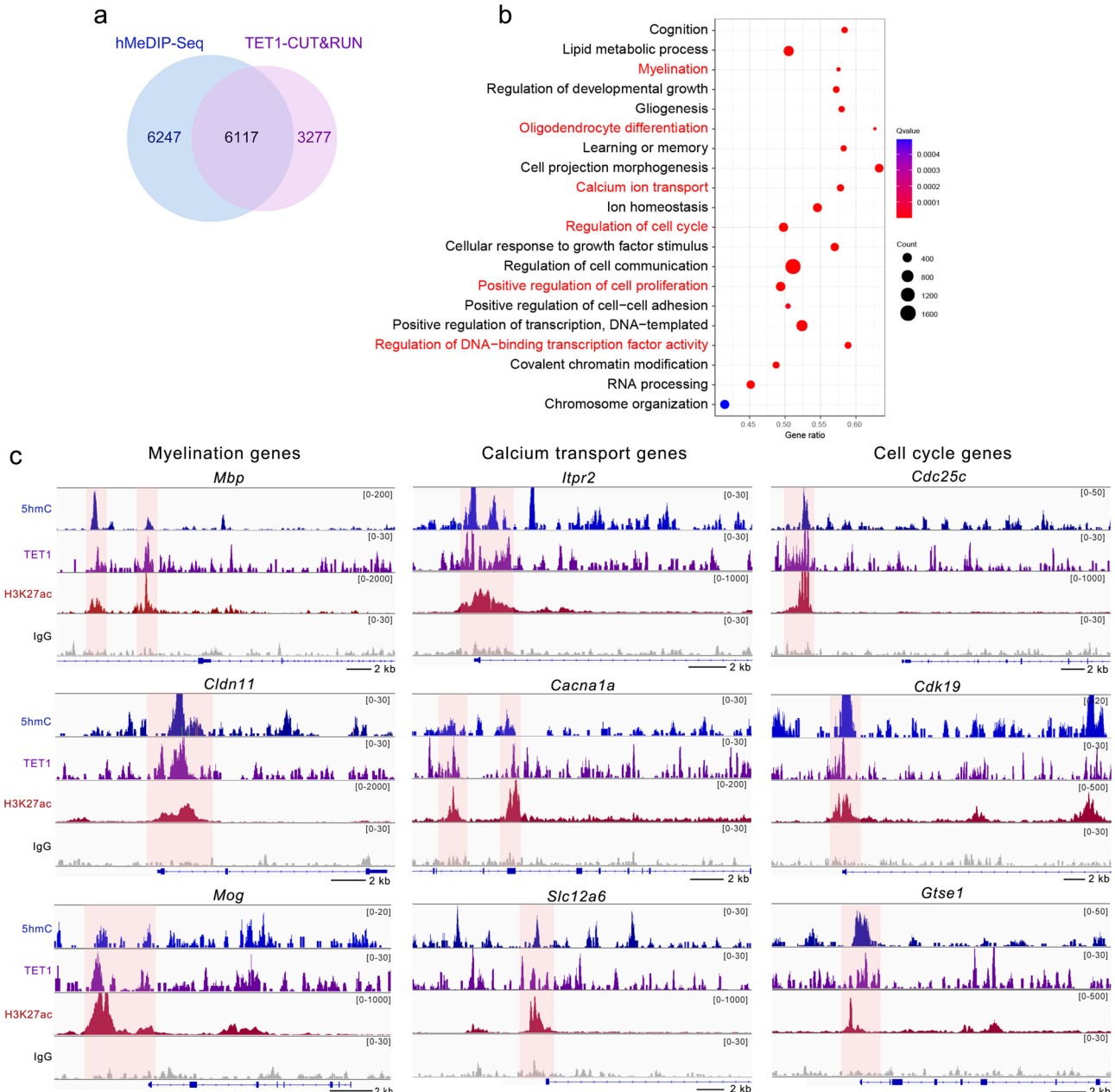

**Fig. 8 Cut & Run-Seq analysis identify TET1 target genes in OPCs. a** Venn diagram comparing hydroxymethylated genes and TET1 binding genes in OPCs. **b** GO analysis for TET1 and 5hmC overlapping target genes in wild-type OPCs. GO terms labeled in red indicate the associated functions of TET1 in our observations. **c** Snapshots comparing 5hmC profiles, TET1 binding profiles, and H3K27ac profiles in representative genes in OPCs.

phase[60]. Consistently, TET1 has been also implicated in stabilizing CCNB1 in trophoblast stem cells and can act as a facilitator of mitotic cell-cycle progression[34,61]. Therefore, it is possible that TET1 may regulate the expression of cell cycle regulators, CCNB1 stability, or both, in OPCs. Although the precise mechanism for cell cycle transition block in *Tet1* cKO OPCs remains unknown, our observations suggest an important role of TET1 in OPC cell-cycle progression.

**TET1-5hmC functions in psychological disorders.** In recent years, TET-5hmC association with psychiatric and cognitive disorders has gained increasing recognition[28–30]. A cohort of new autosomal recessive genes for intellectual disability, including the missense mutations in TET1, has been identified[62]. A previous report using the same *Tet1*loxp line indicated that *Tet1* KO mice

exhibited delayed spatial learning and deficient short-term memory retention in MWM[24]. In contrast, in another study using a different *Tet1* KO line (Δ Exon 4), no significant differences were observed between groups during the training and the probe trials; but short-term memory extinction was impaired in this *Tet1* KO mice[63].

TET1 together with other factors such as ERBB4, BDNF was identified as independent predictors of schizophrenia and serves as a high-risk gene for schizophrenia[31]. Loss of normal PPI of startle, in particular, is widely accepted as an endophenotype of schizophrenia and considered indicative of disrupted sensorimotor gating, a precognitive process to prevent sensory overload and cognitive fragmentation[32,64,65]. Besides, the prodromal cognitive symptoms of schizophrenia often precede the occurrence of psychosis, and the range of cognitive deficits in schizophrenia suggests an overarching alteration in cognitive control, the ability

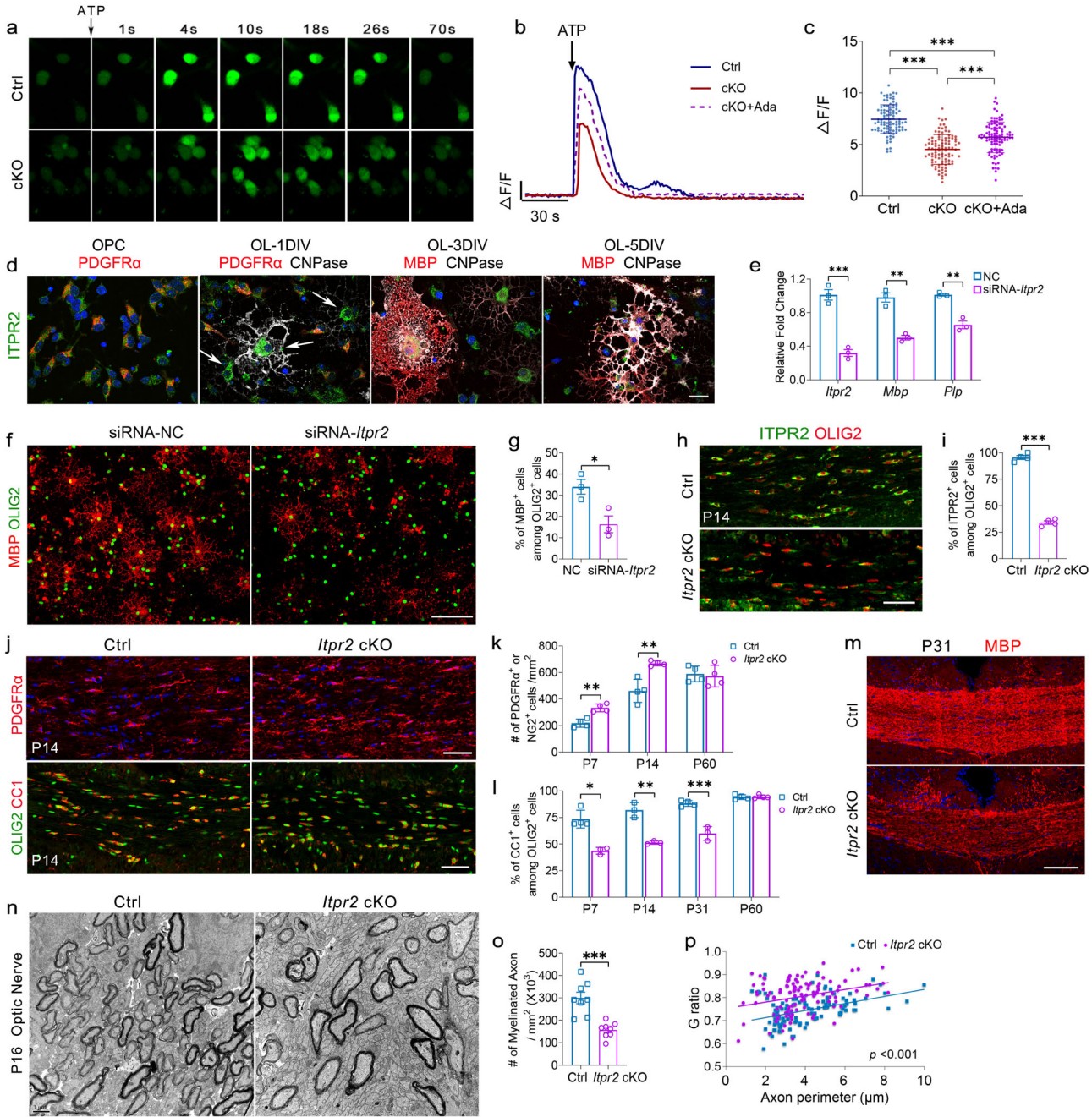

to adjust thoughts or behaviors in order to achieve goals[66–68]. We find that mice with *Tet1* loss in the OL lineage exhibit impaired PPI and working memory deficits, two of the core symptoms related to schizophrenia. Thus, our studies demonstrate a crucial function of TET1-mediated epigenetic modifications in OLs for psychological disorders.

**Locus-specific alterations of DNA hydroxymethylation in OPCs of *Tet1* mutants.** Levels of 5hmC are variable in the promoter and gene body regions[69] and impact gene expression in a cell type-dependent manner[18,70,71]. Our comparison of differentially hydroxymethylated genes with transcriptome profiles indicated that 5hmC signals in gene bodies are more significantly associated with gene expression changes than those in promoter regions in OPCs. This suggests that 5hmC modification by TET1 regulates locus-specific gene expression programs necessary for OPC differentiation. Although the exact mechanism of gene body 5hmC in

regulating gene expression is unclear, there are many observations indicating the association between enriched 5hmC in the gene body and the active transcription in different tissues[36,72–75]. One possibility is that 5mC oxidation relieves a repressive effect on transcription, perhaps by counteracting spurious intragenic anti-sense transcription[76]. Other explanations may include the fact that 5hmC has a destabilizing effect on DNA structure that potentially favors the opening of the double helix by the transcription apparatus[77,78]. Moreover, 5hmC enriched in the gene body cannot be bound by transcriptionally repressive methyl-CpG binding domain (MBD) proteins in vitro[79,80]. These studies suggested that gene body 5hmC levels may regulate the transcription rate by modifying the accessibility of genic chromatin to transcriptional machinery or by inhibiting the binding of repressive methyl-CpG binding proteins (MBDs). How 5hmCs cooperate with other epigenetic regulators for OPC differentiation remains to be determined.

**Fig. 9 Impaired calcium transport in *Tet1* cKO OPCs and depletion of ITPR2, a TET1-5hmC target gene, inhibits developmental myelination.**
**a** Representative serial image was obtained after the addition of 100 μM ATP to OPCs from control or *Tet1* cKO mice. **b** Representative traces of Fluo4 intensity in OPCs from control and *Tet1* cKO mice following application of ATP. Agonist for ITPR2, Ada, is added to rescue impaired calcium rise in *Tet1* cKO mice. **c** Amplitude changes ($\Delta F/F$) after ATP treatment of control and *Tet1* cKO OPCs. ITPR2 agonist Ada is used to rescue calcium rise deficiency in *Tet1* cKO mice. Data were Means ± SD ($n = 97$ cells in Ctrl group from three independent cultures, $n = 99$ cells in cKO group from three independent cultures, $n = 100$ cells in cKO + Ada group from three independent cultures). One-way ANOVA, $F_{(2,293)} = 102.001$, $p < 0.000001$. Tukey's multiple comparisons test: Ctrl vs. cKO: ***, $q = 2.9233$, $p < 0.0000001$, Ctrl vs. cKO + Ada: ***, $q = 1.75171$, $p < 0.0000001$, cKO vs. cKO + Ada: ***, $q = -1.17160$, $p < 0.0000001$. **d** Representative images of ITPR2 immunostaining during OPC differentiation in vitro. Arrows indicate the highest level of ITPR2 in newly formed CNPase$^+$ oligodendrocytes from 1DIV differentiation cultures, which does not appear in other stages. Scale bar, 10 μm. **e** Quantitative real-time PCR identified the efficiency of a siRNA-*Itpr2* duplex in reducing the mRNA level of *Itpr2* in normal OPCs and in repressing the expression of two myelin genes, *Mbp* and *Plp*, in differentiating OLs, respectively. Transfection with nontargeting duplex was used as negative control (NC). Data were Means ± SEM ($n = 3$ transfections). Two-tailed unpaired *t*-test. *Itpr2*: ***, $t = 9.174$, $df = 4$, $p = 0.000784$; *Mbp*: **, $t = 8.083$, $df = 4$, $p = 0.001273$; *Plp*: **, $t = 7.528$, $df = 4$, $p = 0.001667$. **f** Immunostaining for MBP and OLIG2 in OL cultures after transfected with siRNA-*Itpr2* duplex for 4 days. Scale bar, 100 μm. **g** Quantification of the percentage of MBP$^+$ cells among OLIG2$^+$ cells after siRNA transfection. Note the significant decrease of differentiated cells in the siRNA-*Itpr2* transfected group. Data were Means ± SEM ($n = 3$ transfections). Two-tailed unpaired *t*-test, *, $t = 3.372$, $df = 4$, $p = 0.028$ compared to control. **h** Representative double immunostaining for ITPR2 and OLIG2 in P14 corpus callosum from control and *Itpr2* cKO mice. Scale bar, 50 μm. **i** Percentage of ITPR2$^+$ cells among OLIG2$^+$ OLs in P14 corpus callosum of control and *Itpr2* cKO mice. Data were Means ± SEM ($n = 4$ animals in each group). Two-tailed unpaired *t*-test, ***, $t = 25.304$, $df = 6$, $p < 0.00001$ compared to control. **j** Immunostaining for OPC marker PDGFRα and double immunostaining for OL lineage marker OLIG2 and mature OL marker CC1 in corpus callosum from P14 *Itpr2* cKO mice. Scale bar, 50 μm. **k** Quantification of the density of PDGFRα$^+$ cells in P7 and P14 mice revealed a significant reduction in *Itpr2* cKO mice. No difference in density of NG2$^+$ cells was observed in P60 adult mice. Data were Means ± SEM ($n = 4$ animals per group). Two-tailed unpaired *t*-test, P7: **, $t = -5.411$, $df = 6$, $p = 0.002$; P14: **, $t = -4.668$, $df = 6$, $p = 0.003$; P60: $t = 0.336$, $df = 6$, $p = 0.748$ compared to control. **l** Quantification the percentage of CC1$^+$ cells among OLIG2$^+$ cells from P7 to P31 mice revealed significant reduction in *Itpr2* cKO mice. No difference was observed at P60 adult mice. Data were Means ± SEM ($n = 4$ animals in control group and three animals in mutant). P7: *, Mann–Whitney *U*-test, $z = -2.141$, $p = 0.032$. Two-tailed unpaired *t*-test, P14: **, $t = 7.414$, $df = 4$, $p = 0.002$; P31: ***, $t = 7.931$, $df = 5$, $p = 0.0005$; P60: $t = 0.178$, $df = 6$, $p = 0.865$ compared to control. **m** Representative images for MBP staining in corpus callosum from P31 *Itpr2* cKO mice. Scale bar, 50 μm. **n** Representative electron micrographs of P16 optic nerves from control and *Itpr2* cKO mice. Scale bar, 1 μm. **o** Quantification of the number of myelinated axons in defined areas from optic nerves of control and *Itpr2* cKO mice. Data were Means ± SEM ($n = 8$ slides from three animals per group). Two-tailed unpaired *t*-test, ***, $t = 5.035$, $df = 14$, $p = 0.00018$ compared to control. **p** G ratios vs. axonal perimeters for control and *Itpr2* cKO mice reveal a significant difference. Friedman M test, $\chi^2 = 39.035$, $df = 1$, $p_{between\ group} < 0.001$ (>110 myelinating axon counts from three animals each genotype).

Intriguingly, ablation of *Tet1* led to the upregulation of a set of genes in OPCs, indicating that TET1 may also function as a transcriptional repressor. Consistent with our data, inhibition of *Tet1* expression increased the level of a set of genes in ESCs[81,82]. TET1-mediated repression might involve recruitment of the MBD3/NuRD repressor complex, which was shown to co-localize with TET1 in ESCs[83]. TET1 may also coordinate with the SIN3A co-repressor complex, which shows a similar binding profile to TET1 and is required for a subset of TET1-repressed target genes[81,84]. A recent study indicates that TET1-mediated transcriptional repression could channel through the JMJD8 demethylase transcriptional repressor and is independent of TET1 catalytic activity during epiblast differentiation[85]. The mechanisms of locus-specific transcriptional regulation by TET1 during OL development remain to be further defined.

**TET1-5hmC regulates calcium transport to control proper OL differentiation**. Calcium signaling is important for OPC migration, differentiation, and initiation of myelin formation[86–89]. For instance, blocking of voltage-gated Ca$^{2+}$ entry in OPCs inhibits their maturation and myelin formation ability[42]. Similarly, an increase of the resting [Ca$^{2+}$]$_i$ through membrane depolarization facilitates MBP synthesis in OPCs[87]. We found that *Tet1* deletion led to the downregulation of multiple calcium transporter genes in OPCs, which mediated the impaired intracellular store release and extracellular influx in response to a stimulus. Moreover, treatment with a calcium channel agonist reversed the differentiation defects in *Tet1*-deficient OPCs, thus indicating the significance of TET1-5hmC regulated calcium transport in OLs.

We find that ITPR2, an intracellular ER calcium channel that is exclusively expressed in postmitotic OLs[43,44], is one of the TET1-5hmC targets. Expression of ITPR2 is upregulated during a motor learning task[44], indicating the participation of ITPR2$^+$

OLs in myelin plasticity. We find that deletion of *Itpr2* in the OL lineage greatly reduces OPC differentiation, suggesting that ITPR2 is critical for the initiation of myelination. In addition to ITPR2, we have also detected *solute carrier* (*Slc*) gene family members as TET1-5hmC targets. Among these genes, *Slc12a2* is reported responsible for defective myelin regeneration detected in old *Tet1* cKO mice[23]. Therefore, TET1-mediated genome-wide DNA hydroxymethylation may regulate multiple distinct targets including *Itpr2* and *Slc12a2* for optimal myelinogenesis or at different stages of OL lineage cells.

Together, TET1-mediated 5hmC modification, or DNA hydroxymethylation, can modulate the process of oligogenesis and myelinogenesis through at least two critical processes, by fine-tuning cell cycle progression for OPC proliferation and by regulating OL homeostasis e.g., ITPR2-mediated calcium transport, for OL myelination.

## Methods

**Animals, immunohistochemistry, and EM**. All animal experiment protocols were approved by the Animal Care and Use Committee of the Fourth Military Medical University and were conducted in accordance with the guidelines for the care and use of laboratory animals. All mice used in this study were kept under stable 12-h circles of darkness and light in the respective facilities. Room temperature was kept between 20–24 °C and air humidity between 45–65% as documented in daily controls. *Tet1* $^{flox/flox}$ mice[24] and *Tet3* $^{flox/flox}$ mice[18] were crossed with heterozygous *Olig1*-Cre mice[25,90] to generate *Tet1* $^{flox/+}$;*Olig1*Cre$^{+/−}$ mice and *Tet3* $^{flox/+}$;*Olig1*Cre$^{+/−}$ mice, which were then bred with *Tet1* $^{flox/flox}$ or *Tet3* $^{flox/flox}$ mice to produce *Tet1* cKO (*Tet1* $^{flox/flox}$;*Olig1*Cre$^{+/−}$) or *Tet3* cKO (*Tet3* $^{flox/flox}$;*Olig1*Cre$^{+/−}$) offspring, respectively. *NG2CreER* mice[35], R26-EYFP mice (Stock No: 006148), and R26-tdTomato mice (Ai14) (Stock No: 007914) were from Jackson lab. *Itpr2* $^{flox/flox}$ line[91] was provided by the RIKEN BRC (RBRC10293) through the National Bio-Resource Project of the MEXT/AMED, Japan. It was crossed with heterozygous *Olig1*-Cre mice to generated *Itpr2* cKO (*Itpr2* $^{flox/flox}$;*Olig1*Cre$^{+/−}$) as stated above. For tamoxifen treatment of *Tet1* $^{flox/flox}$, *NG2CreER* mice, tamoxifen (Sigma, T5648) was dissolved in corn oil (Sigma, C8267) and injected intraperitoneally at 3 mg/40 g (body weight) per day.

For immunohistochemistry, cryosections (14 μm) of brains were short-fixed 30 min in 4% paraformaldehyde and processed for antigen retrieval. Sections were treated with 10 mM sodium citrate (pH 6.0) at ~90 °C for 10 min in a microwave and cooled down in room temperature. Then sections were washed three times in PBS, blocked in 3% BSA with 0.03% Triton X-100 (blocking buffer) for 1 h at RT, and incubated with primary antibodies in blocking buffer overnight at RT. The next day, sections were washed three times in PBS, incubated with secondary antibodies at RT for 2 h, and then counterstained with DAPI for 5 min. Finally, sections were washed three times in PBS and mounted. Images were taken on an Olympus FV1200 Confocal microscope. We used antibodies against OLIG2 (Millipore AB9610, 1:500), SOX10 (Santa Cruz 17342, 1:50), TET1 (Genetex 124207, 1:100), PDGFRα (Abcam ab61219, 1:500), CC1 (Oncogene Research OP80, 1:200), MBP (Abcam ab7349, 1:500), 5hmC (Active motif 39769, 1:1000), NeuN (Millipore ABN78, 1:800), PV (Abcam ab11427, 1:200), SST (Millipore AB5494, 1:50), VIP (CST 63269, 1:200), ALDH1L1 (Proteintech 17390-1-AP, 1:100), CNPase (Sigma C5922, 1:500), Ki67 (Abcam ab16667, 1:500), GFAP (Millipore mAB360, 1:1000), GST-pi (Abcam ab53943, 1:50), NESTIN (Genetex GTX630201, 1:500), DCX (Millipore ab2253, 1:500), ITPR2 (Millipore ab3000, 1:50), and H3S10P (Cell Signaling Technology 9706, 1:500).

For BrdU pulse labeling in P1 pups, animals were injected intraperitoneally with 100 mg BrdU/kg body weight 2 h prior to sacrifice. For the staining of BrdU, as well as 5hmC, before permeabilization, sections were subjected to DNA denaturation with 2 M hydrochloric acid at 37 °C for 20 min and then neutralized with 0.1 M sodium borate at pH 8.5 for 2 × 10 min. The G3G4 monoclonal antibody (anti-BrdUrd, 1:40) was obtained from the Developmental Studies Hybridoma Bank developed under the auspices of the NICHD and maintained by The University of Iowa.

For TUNEL/OLIG2 double staining, OLIG2 immunostaining were applied after the DeadEnd™ Fluorometric TUNEL System kit (Promega, G2350) to reveal TUNEL positive cells in OL lineage.

For EM, tissues were dissected and fixed in 2% glutaraldehyde + 4% paraformaldehyde in 0.1 M cacodylate buffer (pH 7.2) at 4 °C for 24 h. The brain was coronally sliced at the 50-μm thickness and treated with 2% osmium tetroxide overnight before being subjected to a standard protocol for epoxy resin embedding. Tissues were sectioned at 1 μm and stained with toluidine blue. Ultrathin sections were cut onto copper grids and stained with uranyl acetate before being examined with a JEM-1230 electron microscope (JEOL LTD, Tokyo, Japan), which is equipped with a CCD camera and its application software (832 SC1000, Gatan, Warrendale, PA).

**Western blot assay**. For Western blot assay, whole-cell lysates were prepared from tissue or cultures using RIPA buffer (150 mM NaCl, 50 mM Tris-HCl (pH 7.4), 1% NP40, 1 mM PMSF, 1× Roche complete mini protease inhibitor cocktail, and 1× Pierce phosphatase-inhibitor cocktail). Protein concentrations in centrifugation-clarified cell lysates were measured by the BCA Protein Assay Kit (Pierce) and equal amounts of protein (10 μg) were separated on SDS-PAGE gel and transferred to Hybond PVDF (Amersham Biosciences). For protein blotting, we used antibodies against OLIG2 (Millipore MABN50, 1:1000), MBP (Abcam ab7349, 1:1500), NeuN (Millipore ABN78, 1:2000), ALDH1L1 (Proteintech 17390-1-AP, 1:1000), DCX (Proteintech 13925-1-AP, 1:1000), CCNB1 (Boster BA0766-2, 1:500), β-actin (Proteintech 66009-1,1:5000), GAPDH (Proteintech, 60004-1, 1:5000), and Tubulin (Abbkine A01030, 1:1000). Signals were developed with horseradish peroxidase-conjugated secondary antibodies (Abbkine), followed by an ECL kit (Zeta LIFE). The band intensity was calculated with Tanon5200 imager and quantified with ImageJ 1.52p software. The intensity was normalized to β-actin or Tubulin level expressed as relative fold change against control.

**RNA extraction and qRT-PCR**. Total RNAs were purified from tissues or cell cultures using TRIzol reagent according to the manufacturer's instruction (Invitrogen). For qRT-PCR, RNA was transcribed to cDNA with the PrimeScript™ RT reagent Kit (Perfect Real Time, Takara) and reactions were performed with SYBR® Premix Ex Taq™ (Takara) in CFX96 Touch Real-Time PCR Detection System (Bio-Rad). Relative gene expression was calculated by Bio-Rad CFX Manager and normalized to internal control β-actin. Primer sequences for SybrGreen probes of target genes are listed in Supplementary Table 3.

**Culture of OPCs and immunocytochemistry**. Rodent OPCs were isolated from P6 cortices of animals by immunopanning with antibodies against RAN-2, GalC, and O4 sequentially[92]. Briefly, cerebral hemispheres were diced and digested with papain at 37 °C. Following gentle trituration, cells were resuspended in a panning buffer containing insulin (5 μg/ml) and then incubated at room temperature sequentially on three immunopanning dishes: RAN-2 (ATCC #TIB-119, 1:5 hybridoma supernatant), anti-GalC (1:5 hybridoma supernatant)[93], and O4 (1:5 hybridoma supernatant)[94]. O4+ GalC− OPCs were released from the final panning dish with trypsin (Sigma). To induce the differentiation of OPCs, mitogens in the culture medium (FGF and PDGFaa) were replaced with T3, NAC, and CNTF.

For immunocytochemistry, cell cultures were fixed in 4% PFA. After Triton X-100 permeabilization for 15 min, samples were incubated with primary antibody for 1 h at room temperature followed by the fluorescent secondary antibody for

another hour. Cells were then counterstained with DAPI and visualized with a confocal microscope. Experiments were replicated using cells from three different primary cultures.

**Flow cytometric analysis of cell cycle with propidium iodide DNA staining**. PI staining for flow cytometry was performed according to the user manual of DNA Content Quantitation Assay (Cell Cycle) from Solarbio (#CA1510). Briefly, OPCs from control or *Tet1* cKO mice were harvested, washed in PBS, and fixed in cold 70% ethanol for 30 min at 4 °C. After washing twice in PBS, cells were treated with RNase and then stained with PI. With guava easyCyte6HT (Millipore), the forward scatter (FS) and side scatter (SS) were measured to identify single cells. For analysis, ModFit LT 5.0 software was used to make the PI histogram plot. Experiments were replicated three times.

**Lysolecithin-induced demyelinating injury**. Lysolecithin-induced demyelination was carried out in the corpus callosum of 8-week-old mice. Anesthesia was induced and maintained by peritoneal injection of a mixture of ketamine (90 mg/kg) and xylazine (10 mg/kg). The skull was exposed, and a hole was cut into the cranium. Focal demyelinating lesions were induced by stereotaxic injection of 0.8 μl 1% lysolecithin solution (L-a-lysophosphatidylcholine, Sigma L4129) into the corpus callosum at coordinates: 0.8 mm lateral, 0.8 mm rostral to bregma, 1.2 mm deep to brain surface) using a glass capillary connected to a 10 μl Hamilton syringe. Animals were left to recover in a warm chamber before being returned to their housing cages. LPC-induced injuries were conducted in a genotype-blinded manner.

**Electrophysiology**. Analyzing the CAP was performed according to previous protocols[27,95]. *Tet1* cKO and control littermates were killed by cervical dislocation and then decapitated. Optic nerves were dissected free and cut between the orbit and the optic chiasm in the standard aCSF containing (in mM): NaCl 126, KCl 3.0, CaCl₂ 2.0, MgCl₂ 2.0, NaH₂PO4 1.2, NaHCO₃ 26, and glucose 10 at 37 °C. Following dissection, optic nerves were equilibrated in aCSF for at least 30 min with constant aeration (95% O₂/5% CO₂). Then the nerve was gently placed in a customized perfusion chamber, maintained at 37 °C, and perfused with aCSF at 2–3 ml/min speed. Suction electrodes backfilled with aCSF were used for stimulation and recording. One electrode was attached to the rostral end of the nerve for stimulation and the second suction electrode was attached to the caudal end of the nerve to record the CAP, thus all recordings were orthodromic. Stimulus pulse strength (100-μs duration, SS-201J Isolator constant current output with SEN-7203 stimulator, Nihon Kohden, Japan) was adjusted to evoke the maximum CAP possible and then increased another 25% (i.e., supramaximal stimulation). During an experiment, the supramaximal CAP was elicited every 10 s and repeat ten times. The signal was amplified 100 × AC membrane potential (100 mV/mV) by a Multiclamp700B amplifier, filtered at 10 kHz, and acquired at 10 kHz by Clampex 10.6 software (Axon Instrument, Molecular Devices, USA). The average CAP amplitude and area were measured in Clampfit 10.6 software (Molecular Devices, USA) offline and performed blind to genotype. Image drawing and statistical analysis were performed in GraphPad Prism 8.

**Behavior test**

*Startle response/PPI tests*. A startle reflex measurement system (Labmaze ZS-ZJT, Beijing) was used to test the startle response and PPI for P42 mice. XeyeStartle software was used to record and analysis the data. Throughout the session, the startle system delivered a constant background white noise of 68 dB. The startle response was recorded for 300 ms (measuring the response every 1 ms) with the onset of the stimulus and a startle response was defined as the peak response during the 300 ms period.

The acoustic startle begins by placing a mouse in the undisturbed chamber for 5 min. The test consists of ten 20-ms bursts of white noise varied in level from 65–125 dB sound stimuli in steps of 5 dB, plus ten no-stimulus trials. The order of these stimuli was randomized, and the duration of intertrial intervals was 15 s. The PPI test session began with a 5-min acclimation period followed by three consecutive blocks of test trials. Block 1 and 3 consisted of six startle stimulus-alone trials. Block 2 contained ten startle stimulus-alone trials, ten prepulse + startle trials per prepulse intensity, and ten no-stimulus trials. Three combinations of prepulse and startle stimuli (70–120, 76–120, and 82–120 dB) were employed. Trials were presented in a pseudorandom order, ensuring that each trial was presented ten times and that no two consecutive trials were identical. Intertrial intervals ranged from 30 to 45 s. Basal startle amplitude was determined as the mean amplitude of the ten startle stimulus-alone trials. PPI was calculated according to the formula: 100*[1 − (PPx/P120)]%, in which PPx means the ten PPI trials (PP70, PP76, or PP82 and P120 was the basal startle amplitude).

*Morris water maze*. The MWM was conducted as described in ref. [33] with minor modifications. A white plastic tank 120 cm in diameter was kept in a fixed position and filled with 22 °C water, which was made opaque with milk. A 10 cm platform was submerged 1 cm below the surface of opaque water and located in the center of one of the four virtually divided quadrants. All animal activities were automatically recorded and measured by Smart 3.0 software. P90 mice were tested for MWM.

The swim training consisted of 5 days of trials, during each day mice were released from four random locations around the edge of the tank with an intertrial interval of about 30 min and they were allowed to freely swim for a maximum of 60 s or guided to the platform. Afterward, mice were allowed to stay on the platform for 15 s. A probe trial was performed 24 h after the last day of training. During the probe trial, mice were allowed to swim in the pool without the escape platform for 60 s. The performance was expressed as the percentage of time spent in each quadrant of the MWM and swim distance in the target quadrant, which were automatically recorded. Moreover, the latency to reach the platform position (using 10 cm diameter) and the number of crossings through the position were manually recorded.

*RNA-seq and data analysis.* RNA-seq were performed by RiboBio Co., Ltd. (Guangzhou, China). Briefly, libraries were prepared using Illumina RNA-Seq Preparation Kit (TruSeq RNA Sample Prep Kit) and sequenced by HiSeq 3000 sequencer. RNA-seq reads were mapped to mm10 using TopHat 2.1.1 with settings of "read mismatches=2" and "read gap length=2" (http://ccb.jhu.edu/software/tophat/index.shtml). TopHat output data were then analyzed by DEGseq (1.39.0) package to compare the changes of gene expression between *Tet1* cKO and control, based on the calculated RPKM values for known transcripts in mouse genome reference. Heatmap of gene differential expression was generated using R Package (http://www.r-project.org). GO analysis was performed using ToppGene Suite (http://toppgene.cchmc.org) and GSEA 4.0.1 (http://www.broadinstitute.org/gsea/index.jsp).

*hMeDIP-sequencing analysis.* hMeDIP-sequencing service was provided by Kang-Chen Biotech (Shanghai, China). hMeDIP-sequencing library preparation was performed according to a previous study[96] with minor modifications. Genomic DNA was sonicated to ~200–800 bp with a Bioruptor sonicator (Diagenode). About 800 ng of sonicated DNA was end-repaired, A-tailed, and ligated to single-end adapters following the standard Illumina genomic DNA protocol (FC-102-1002, Illumina). After agarose size-selection to remove unligated adapters, the adapter-ligated DNA was used for IP with a mouse monoclonal anti-5hmC antibody (Diagenode, C15200200). For this, DNA was heat-denatured at 94 °C for 10 min, rapidly cooled on ice, and immunoprecipitated with 1 μl primary antibody overnight at 4 °C with rocking agitation in 400 μl IP buffer (0.5% BSA in PBS). A nonspecific IgG control was included to normalized DNA modification enrichment. To recover the immunoprecipitated DNA fragments, 20 μl of magnetic beads were added and incubated for an additional 2 h at 4 °C with agitation. After IP, a total of five washes were performed with an ice-cold IP buffer. Washed beads were resuspended in TE buffer with 0.25% SDS and 0.25 mg/mL proteinase K for 2 h at 65 °C and then allowed to cool down to room temperature. DNA was then purified using Qiagen MinElute columns and eluted in 16 μl EB (Qiagen). Fourteen cycles of PCR were performed on 5 μl of the immunoprecipitated DNA using the single-end Illumina PCR primers. The resulting products were purified with Qiagen MinElute columns, after which a final size selection (300–1000 bp) was performed by electrophoresis in 2% agarose. Libraries were quality controlled by Agilent 2100 Bioanalyzer.

For sequencing, the library was denatured with 0.1 M NaOH to generate single-stranded DNA molecules and loaded onto channels of the flow cell at 8 pM concentrations, amplified in situ using TruSeq Rapid SRCluster Kit (GD-402-4001, Illumina). Sequencing was carried out by running 150 cycles on Illumina HiSeq 2500 using TruSeq Rapid SBS Kit (FC-402-4001, Illumina) according to the manufacturer's instructions.

After sequencing images generated, the stages of image analysis and base calling were performed using Off-Line Base caller software (OLB V1.8). After passing the Solexa CHASTITY quality filter, the clean reads were aligned to mm10 using BOWTIE software (V2.1.0). Aligned reads were used for peak calling, both mRNA and LncRNA associated hMeDIP enriched regions (peaks) with statistically significant were identified for each sample, using a $q$-value threshold of $10^{-4}$ by MACS v2 (http://liulab.dfci.harvard.edu/MACS). Both mRNA and LncRNA associated hMeDIP enriched regions (peaks) were annotated by the nearest gene using the newest UCSC RefSeq database. Differentially hydroxymethylated regions (DhMRs) between two groups with statistically significant were identified by diffReps (Cut-off: $log_2FC = 1.0$, $p$ value $= 10^{-4}$). DhMRs were annotated by the nearest gene using the UCSC RefSeq and database of multiple databases integration.

*qPCR assay for loci-specific 5hmC modification.* To further confirm the loci-specific hydroxymethylation identified in hMeDIP-sequencing, we tested the 5hmC level by qPCR with BisulPlus™ Loci 5mC and 5hmC Detection PCR Kit (EpiGentek, #P-1067). Briefly, 200 ng genomic DNA from OPCs were subjected to bisulfite modification according to the protocol and cleaned up for enzyme conversion. During DNA bisulfite treatment, unmodified cytosine (C) is converted to uracil and will be read as T in the PCR; 5mC remains the same and 5hmC forms cytosine 5-methylenesulfonate (CMS). Further treatment with specific APOBEC deaminase will convert 5mC to thymine but not affect CMS. The bisulfite-enzyme converted DNA can then be used for qPCR of loci-specific detection of 5hmC and ~10 ng DNA template was used in each PCR reaction, under standard thermal conditions (7 min hot start, 95 °C denaturation for 10 s, 53.5 °C annealing temp, and 72 °C

extension for 50 cycles, 72 °C final extension for 1 min) in a 20 μl reaction. During the PCR, CMS will still be read as C so that 5hmC can be discriminated not only from C, 5mC but also other modified cytosines such as 5fC and 5caC. The primer's design follows the same criteria for Bisulfite conversion-based methylation-specific PCR (MSP)[97] and selected with MethPrimer (http://www.urogene.org/methprimer/index.html). Primers are listed in Supplementary Table 1. Two pairs of primers are needed, one is specific for modified and hydroxymethylated DNA, and the other for modified and unhydroxymethylated DNA. We calculate 5hmC abundance using the following equation:

$$\text{Relative 5hmC abundane} = 2^{[(Ct_{OPC\ methylated}-Ct_{OPC\ unmethylated})-(Ct_{NPC\ methylated}-Ct_{NPC\ unmethylated})]}$$

*Cut & Run-seq and data processing.* Cut & Run-seq was performed following the published protocol[98]. Two million OPCs were harvested by centrifugation (600xg, 3 min) and cells were washed twice in 1 ml Wash Buffer (20 mM HEPES-KOH pH 7.5; 150 mM NaCl; 0.5 mM Spermidine; 1x Roche cOmplete™), followed by the addition of 10 μl pre-activated concanavalin A coated magnetic beads (Bangs Laboratories-BP531). Tubes were held on a mixing platform at RT for 10 min, then placed on a magnet stand and pulled off the liquid. Cells were resuspended in 100 μl cold Antibody Buffer and 1 μl antibody (TET1, Active Motif #61943) was added for incubating on nutator overnight at 4 °C. Beads with cells were washed twice in 1 ml Digitonin Buffer (20 mM HEPES-KOH pH 7.5; 150 mM NaCl; 0.5 mM Spermidine; 1x Roche cOmplete™; 0.05% digitonin), and resuspended in 50 μl Digitonin Buffer with the addition of 2.5 μl CUTANA pAG-MNase (20x stock). After 10 min incubation at RT, samples were washed twice in 1 ml Digitonin Buffer and changed to 50 μl cold Digitonin Buffer. Then tubes were placed in a metal block in ice water and quickly mixed with 100 mM CaCl₂ to a final concentration of 2 mM. The reaction was quenched by the addition of a 50 μl Stop Buffer. Cleaved fragments were liberated into the supernatant by incubating the tube for 30 min at 37 °C. DNA fragments were purified by Qiagen MinElute PCR Purification Kit from the supernatant and used for the construction of sequencing libraries. Libraries were prepared for Illumina sequencing with Library Prep for Cut & Run with NEBNext® Ultra™ II DNA Library Prep Kit for Illumina® (E7645) and were sequenced by NovaSeq PE150.

Cut & Run-seq reads in FASTQ format were first subjected to quality control to assess the need for trimming of adapter sequences or bad quality segments. The programs used in these steps were FastQC v0.11.7, Trim Galore! v0.4.2 and cutadapt v1.9.1. The trimmed reads were aligned to the reference rat genome version Rn5 with the program BOWTIE v2.3.4.1. Aligned reads were stripped of duplicate reads with the program sambamba v0.6.8. Peaks were called using the program MACS v2.1.2 with the narrow and broad peaks mode for Cut & Run-seq.

*ER [Ca²⁺] measurement.* To monitor internal ER store Ca²⁺ level, a low-affinity calcium indicator mag-Fluo4-AM kit was used according to the manufacturer's instructions (GENMED Scientifics INC., USA, GMS10267)[99]. Briefly, OPCs grown on the poly-L-lysine-coated 96-well black-walled plate were washed with intracellular-like medium (ICM) with EGTA and loaded with mag-Fluo4-AM at 37 °C for 45 min. Then cells were washed briefly with ICM and permeabilized by 2 min exposure to β-escin. After washing with ICM, fluorescence (excitation 488 nm, emission 515 nm) was recorded at FlexStation 3 spectrofluorometer (Molecular Devices). The relative fluorescence intensity was determined with the following equation, $[Ca^{2+}] = K_d \times (F - F_{min}) \div (F_{max} - F)$, and compared to OPCs from control mice. Dye calibration was achieved by obtaining $F_{min}$ from blank wells with no cells as a baseline value, and $F_{max}$ from cells loaded with high $[Ca^{2+}]$ as saturated value.

*[Ca²⁺]ᵢ imaging.* Fluo4 was used to measure $[Ca^{2+}]_i$ changes. To load cells with Ca²⁺ probe, cultures were incubated in standard aCSF containing (in mM): NaCl 125, KCl 3.0, CaCl₂ 2.0, MgCl₂ 2.0, NaH₂PO₄ 1.25, NaHCO₃ 26 and glucose 20, or calcium-free aCSF (NaCl 125, KCl 3.0, MgCl2 2.0, MgSO4 2.0, NaH2PO4 1.25, NaHCO3 26, glucose 20, HEPES 10, and EGTA 10 (pH 7.4), supplemented with 0.03% Pluronic F-127 and 0.6 μM cell-permeable form of indicator (Fluo4-AM, Invitrogen) for 20 min at 37 °C. After loading, cells were transferred into a recording chamber on the stage of Olympus FV1000 confocal inverted microscope equipped with phase-contrast optics. Measurements started after at least 15 min storage in aCSF to ensure deesterification of indicators. Fluo4 was excited with a 488 nm laser and emitted light was collected at 515 nm. A series of sections were collected every 500 ms, at 500 ms intervals, for 180 s in total. Pharmacological agents were directly added in aCSF: ATP (Sigma, 100 μM) or Bay K 8644 (10 μM), 30 s after the first image was taken. The images were analyzed using the Olympus FV10-ASW 4.1 software. Cell bodies were selected as regions of interest (ROI) and normalized changes of Fluo4 fluorescence intensities were calculated as $\Delta F/F = (F - F_0)/F_0$ ($F$ fluorescence intensity, $F_0$ baseline intensity). Data were expressed as means ± standard deviation (SD). "$n$" represents the number of responding cells. Experiments were performed three times from different cultures and the results were pooled together for analysis. $n > 97$ in ATP treatment group and $n > 130$ in Bay K 8644 group.

*Duplex siRNA transfection.* For in vitro *Itpr2* knockdown, purified OPCs were transfected with 50 nM duplex siRNA against *Itpr2* (sense, 5′ GGUACCAGCUAA ACCUCUUTT 3′; anti-sense, 5′ AAGAGGUUUAGCUGGUACCTT 3′) or control

nontargeting siRNA (Genepharma, Shanghai) using Lipofectamine RNAiMAX (Invitrogen). Six hours after transfection, the cultures were changed to a differentiation medium. Two or four days later, cultures were harvested for qRT-PCR assay or immunocytochemistry as indicated.

**Statistics and reproducibility.** All statistical analyses were performed in Prism v.8.0 (GraphPad Software, Inc.) and SPSS v.21.0 (IBM Inc.). The normality test was performed by the Shapiro–Wilk test and the homogeneity of variance test was performed by Levene's test. All data met normality and homogeneity of variance were compared using a two-tailed unpaired $t$-test and RMANOVA. Data sets that were normal distribution but not homogeneity of variance were compared using a two-tailed unpaired separate variance estimation $t$-test. Data that didn't meet normal distribution were analyzed with Mann–Whitney $U$-test or Friedman M test. Numerical values were described using mean ± SEM and presented as bar graphs. Significance is denoted as *$p < 0.05$, **$p < 0.01$, or ***$p < 0.001$ in the figures.

For all quantifications in immunostaining and EM, $n = 3$–5 animals, independent cultures, or transfections were examined as indicated in the figure legends (3–6 images were analyzed and averaged per mouse or culture, for each staining). All statistical details for each graph can be found in the figure legends. qRT-PCR on cell cultures have been replicated three times in independent experiments.

**Reporting Summary.** Further information on research design is available in the Nature Research Reporting Summary linked to this article.

## Data availability

The transcriptome, DNA hydroxymethylation profiling, and Cut & Run-seq data generated in this study have been deposited in the Gene Expression Omnibus database under accession code GSE122838. Previously published and deposited data for hMe-Seal-seq of WT NSCs were extracted from GSE65994 and ChIP-seq of H3K27ac were extracted from GSE42454. The data supporting this study are available in the Article, Supplementary Information, or available from the corresponding authors upon reasonable requests. Source data are provided with this paper.

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

## Acknowledgements

This work was supported by the National Natural Science Foundation of China (Grant number: 82071271 and 31571050 to X.Z., 81730035 to S.W.). Q.R.L. was supported by the Cincinnati Children's Hospital Medical Foundation and CancerFree KIDs foundation. The authors would like to thank Dr. Fangfang Liu, Dr. Bo Zhao, Junjun Kang, Ke Fang, and Haifeng Zhang for great technical support; Dr. Natalie Lai Man Wu for GSEA analysis; Dr. Weidong Tian, Dr. Feng Zhang, Dr. Yazhou Wang, Dr. Ceng Lou, Dr. Rougang Xie, Dr. Jinxiang Dai, and Dr. Yaqi Deng for helpful suggestions; and Dr. Guoliang Xu for *Tet1*loxp lines.

## Author contributions

Data acquisition, M.Z., J.W., K.Z., G.L., Y.L., K.R., D.X., J.X. and X.G.; Experimental design, validation, data analysis, visualization, and original draft, M.Z., W.W., L.X., H.M., W.J. and X.Z.; Conceptualization, writing, revisions, and editing, M.Z., K.B., S.W., Q.R.L. and X.Z.; Funding acquisition and supervision, S.W., Q.R.L. and X.Z.; Resources, K.M.

## Competing interests

The authors declare no competing interests.
