## [Peer Review File · Nature Communications]

Reviewers' Comments:

Reviewer #1:

Remarks to the Author:

Zhang et al. showed that a genome-wide shift in 5-hydroxy-methyl-cytosine (5hmC) landscape during NPC to OPC (so OL lineage specification) and hypothesized that such changes are caused by TET1, an abundant member of TET family in OPCs. They characterized Olig1-Cre TET1 cKO mice and revealed that TET1 functions are required for early OPC cell proliferation control and OL lineage progression, as well as remyelination. However, the cKO phenotypes were only prominent at young ages. This study further analyzed the potential targets of TET1 using RNA-seq and hMeDIP-seq approaches and proposed numerous calcium channels and intracellular transporters, such as type II IP3R. Interestingly, cKO OPCs exhibit altered intracellular calcium dynamics in response to some VOCC agonists. Also, the authors characterized Olig1-Cre-mediated Itr2 cKO, showing similar myelin defects found in Tet1 cKO.

This study delivers important mechanistic insights into how DNA demethylation (specifically, TET1-dependent deoxygenation of methylcytosine) regulates early OL development and myelin repair, and their potential relevant mechanisms.

However, given the importance of these scientific findings, the following points should be addressed, and the results interpreted accordingly.

♣ Page 3 - 4 in Results: This study compared 5hmC distribution between NPCs and OPCs in Fig.1. The results from this comparison should be interpreted as changes that occur during from NPC to OPC transition. The most appropriate term describing this process may be during 'OL lineage specification' as the authors used in introduction (page 3 line22). However, in this study, this is confused by mixing up terms 'OL lineage differentiation (page 3 line 35)', 'OL lineage transition (page 3, line 36)', or OL lineage progression (page 4, line 19), all of which potentially misleading or confusing future authors. The latter three refer to "OPC to mature oligodendrocyte transition".

♣ Page 4, line 2 - 3: Although the OPC isolation was described in methods section, it may help readers' understanding of the nature of OPCs by stating, "OPCs isolated via immunopanning from P6 mice".

♣ Consider using genetic designations more consistently or differently: Tet1f/f (page 4 line 25) or Tet1flox/flox (line 27). Olig1Cre +/- > Olig1cre/+

♣ One of the most critical issues in this study is whether Olig1-Cre is reasonably specific to OL lineage. Given some previous publications (Jakovcevski et al., 2005; De Biase et al., 2011; Qi et al., 2016), Olig1 may also be expressed neural progenitors during embryonic stages. Provide low power fluorescence and high power confocal brain images of Olig1-Cre; R26-tdTomato (Ai14) or Olig1-Cre crossed with any widely-used cre reporters.

♣ Supplementary Figure 5 / Discussion

In discussion, the authors stated, "Tet1 cKO in the oligodendrocyte lineage exhibit impaired PPI and working memory deficits, the key behavior phenotypes related to schizophrenia. Thus, our studies demonstrate a crucial function of TET1-mediated epigenetic modifications in oligodendrocytes for schizophrenia and related brain disorders."

- What is age of the mice used for this behavior test? It is hard to find the information of mice ages for the behavior test. If adult mice were used, how about OL and myelin phenotypes for those ages? Are these behavioral abnormalities related to myelin phenotypes in Tet1 cKO mice? At the ages of those mice, did authors see myelin abnormalities in cKO mice?

- In order to support the claim that oligodendrocyte (or OL lineage) -specific TET1 is critical PPI and working memory, it will be important to confirm that Olig1-Cre is really specific to OL lineage. Again, images from Olig1-Cre; R26-tdTomato (Ai14) for ventral brain, hippocampus and cortex should be provided. Otherwise, the interpretation must be more carefully stated and modified.

♣ Page 7 lines 3-5, Fig.3 f -g:

Concerning the interpretation of the flow cytometry of PI-labeled OPCs: What is the block mechanism for G2/M to G1 transition? Is there any reference that described a block of G2/M to G1 transition? Alternatively, is there the possibility that this is caused by G2 arrest? It would be great if the authors use phospho-H3 (an M-phase marker) to distinguish G2 arrest from other mechanisms.

- ♣ Are the OPC cell cycle abnormalities seen at older ages? At least, the authors need to provide data of the density of BrdU+Olig2+ cells at different ages.
- ♣ How does Tet1 deletion affect numbers of OPCs only up to P4? Is this short-effect related to compensation by other TET or TET1 expression declines with age (from P4 to P30)?
- ♣ Can in vitro OPC culture (cKO and control) repeat similar results (i.e., cell cycle block)?
- ♣ Page 7 line 23: Correct the designation of Tet1 iKO: NG2-CreERT: Tet1flox/flox (do not colon) > NG2-CreERTM; Tet1flox/flox (CreERTTM and use semicolon)
- ♣ The results of Fig.3 h and I (impaired OPC to OL transition in vitro) do not seem to fit the rest of Fig.3 (defective cell cycle). Shouldn't these results be moved to Fig.4? Can the decreased differentiation be quantified as the percentage of MBP+ cells/Olig2+ cells per a unit area?
- ♣ In order to test the possibility that the functional consequences of Tet1 deletion are compensated by other Tet members in Tet1 cKO (Fig. 2e), observing NG2-CreER Tet1 iKO at older ages will be highly significant. How about testing the same experiments of Figure 4 for adult ages (such as tamoxifen injection at P30 and observation at P70). The author should include Cre reporter to accurately assess the OPC fate and degree of OL lineage progression.
- ♣ Unlike P1 brain of Tet1 cKO, Tet iKO exhibited that number of OPCs increased and OPC proliferation increased, indicating different patterns of OPC proliferation compared with P1 OPCs. Explain.
- ♣ Fig 5b -d: There are many Tet1+Olig2- cells after injury. What are those cells? The effectiveness of Tet1 deletion by NG2-CreER and the tamoxifen protocol?
- ♣ This study did not address whether 5hmC levels changes in vivo, or genome-wide 5hmC landscape changes during OPC-to-OL transition (truly OL lineage progression). Can the authors address these questions? Alternatively, the relevant discussion will be helpful to improve this manuscript.
- ♣ In Fig.8 a-d: in order to separate Ca influx from Ca release-dependent calcium traces, can the authors use different agonists or Ca-free extracellular conditions in calcium imaging? What is the size of internal Ca store in cKO OPCs?
- ♣ In Fig. 8i and j: IP3R (type II) is also known to be expressed in astrocytes. The authors should show astroglial Itpr2 levels remain unaltered.
- ♣ In Fig. 8k – m: Can the authors add one or two more older ages regarding numbers of OPCs and OLs in control and Itpr2 cKO?
- ♣ The authors stated in discussion.. "Despite early developmental defects, we noticed that the developmental myelin deficiencies recovered in adult Tet1 cKO mice, which might be due to the expansion of OL numbers that escaped from Cre-mediated Tet1 depletion. " Can this be tested by TET1 co-immunostaining with CC1 at the adult ages.
- ♣ In the title and abstracts: "TET1-mediated oligodendrocyte homeostasis ..." What do the authors mean by 'oligodendrocyte homeostasis'? The meaning of homeostasis implies any intracellular mechanisms?

References)

Jakovcevski I, Zecevic N. Olig transcription factors are expressed in oligodendrocyte and neuronal cells in human fetal CNS. J Neurosci. 2005;25(44):10064-10073. doi:10.1523/JNEUROSCI.2324-

05.2005

Qi Q, Zhang Y, Shen L, et al. Olig1 expression pattern in neural cells during rat spinal cord development. *Neuropsychiatr Dis Treat.* 2016;12:909-916. Published 2016 Apr 18.
doi:10.2147/NDT.S99257

Reviewer #2:

Remarks to the Author:

The manuscript by Zhang et al revealed TET1 mediated DNA hydroxymethylation is important and necessary for CNS Myelination and Remyelination. Their results showed the olig1-Cre mediated Tet1 depletion mice exhibit myelination deficits at the early postnatal stage, thus induce schizophrenia relevant behaviors, like impaired PPI and working memory; the underlined mechanism has also been experimentally explored as the OPC cell-cycle is blocked at the transition from G2/M to G1 phase in Tet1 cKO mice. The impaired remyelination phenotype in old mice brain has also been observed upon Tet1 loss, which is consistent with the recent report by Moyon et al. The author identifies several calcium transport related genes as major Tet1 targets which contribute to deficient OL differentiation, especially *Itp2*, and ablation *Itp2* in mice will lead to reduced OL development. Overall, this work illuminates a novel and unique role of TET1, but not TET2 or TET3, in OPC and OL differentiation, the underlined mechanism study is solid and clear, their findings are significant and meaningful.

Following are my main comments:

1, According to the authors' previous report (Zhao et al, 2014), Zhao et al showed all TETs members are expressed in OLs and required for the differentiation of OLs in vitro. Here, the authors identified only TET1 as an impact factor in OL differentiation, could the authors give a proper explanation on the different results?

2, Based on a recent report (Moyon et al, 2019), Tet1 cKO has little effect on OPC to OL differentiation in the young mice brain (like P14), the two results (including CC1, Olig2, MBP staining) are inconsistent. How can the authors explain this, and which result is more reliable? While Moyon et al reported several members of the solute carrier (Slc) gene family, including *Slc12a2*, are responsible for defective myelin regeneration detected in old Tet1 cKO mice, so which one is more important? The solute carrier (Slc) gene family or calcium transport related genes, like *Itp2*, reported by the authors in this study? The authors could compare the difference and discuss it. Are the solute carrier (Slc) gene family also found in the gene list of downregulated and low 5hmc enriched overlap in Tet1 cKO mice in this study?

3, The authors identified more gene body regions associated with OL lineage progression when comparing the hMeDIP-seq data in OPC and NPC. Later, they also focus on genes that show both downregulated mRNA expression and decreased gene body 5hmc levels in Tet1 cKO mice OPC. Could the authors try to explain how gene body 5hmc affect gene expression? In their discussion, they mentioned histone H3K27me3/H3K4me3 modification may contribute to this, but this "5hmc accumulates in bivalent domains" mechanism has little relationship with the gene body regions and is not convincing. Besides, while the authors showed TET1 has dual effect on gene expression: activation or repression, why only the active genes are related to OL differentiation? The authors should try to illuminate them in the discussion.

4, While the authors show the ablation of Tet1 results in OPC cell-cycle progression, they performed flow cytometry and found increasing percentages of cells in S and G2/M phase, but decreasing in G1 phase. To make their conclusion more persuasive, the authors could perform cell cycle synchronization assay to clearly and directly show the defects in OPC cell-cycle. Besides, the authors show *Ccng2*, *Cdca3*, *Ccna2*, *Ccnb2*, and *Cdc25b*, which encode factors involved in cell-cycle regulation are enriched by 5hmc modification, may contribute to cell cycle blockade. While TET1 is also critical to the stability of cyclin B1 and acts as a facilitator of mitotic cell-cycle progression, which pathway is the major contribution to cell cycle inhibition: TET1-mediated 5hmc modification activates multiple gene expression or influence the cyclin B1 stability?

5, According to Supplementary Movie 1 and 2, before adding ATP and monitoring the [Ca²⁺] i

increase, the Tet1 cKO OPCs already have a very high fluorescence signal. The high background could cover the real calcium rise. The authors should repeat this assay using a similar initial fluorescence signal in control and Tet1 cKO OPCs.

6, While *Itpr2* cKO mice indeed impair the calcium release and inhibit OL differentiation, these results couldn't demonstrate *Itpr2* is a key downstream target of TET1-mediated 5hmC modifications which is responsible for impaired OL differentiation in Tet1 cKO mice. The authors should perform the rescue assay to put back *Itpr2* in Tet1 cKO OL cultures to monitor the differentiation ability.

7, In the discussion, the authors claim "Despite early developmental defects, we noticed that the developmental myelin deficiencies recovered in adult Tet1 cKO mice, which might be due to the expansion of OL numbers that escaped from Cre-mediated Tet1 depletion", their explanation is not convincing, as the authors' previous report (Zhao et al, 2014), as well as a recent study (Moyon et al, 2019), show TET1 is significantly downregulated in old mice' OLs. The authors could easily detect the Tet1 expression in adult Tet1 cKO mice to verify this possibility, otherwise, the authors should try to give proper explanations.

Reviewer #3:

Remarks to the Author:

Ten-eleven translocation (TET) proteins-mediated cytosine hydroxylation has been shown to play important roles in neurodevelopment and diseases. However, their role in oligodendrocyte homeostasis, myelination and remyelination is unclear. In this manuscript, the authors provide the data suggesting that 5hmC and Tet proteins, particularly TET1, could play roles during oligodendrocyte-progenitor (OPC) differentiation. The topic is certainly interesting and important, however, the data presented in this manuscript are preliminary and lack molecular mechanistic insights. Below are some specific comments:

1. Figure 1-the authors used 5hmC antibody for IP followed by high-throughput sequencing to profile 5hmC dynamics during OL lineage differentiation. It is well known that 5hmC antibody IP-based approach has bias towards heavily methylated regions. The authors need to use some other methods to validate and confirm their findings.
2. Figure 2 and 3 suggest that the loss of TET1 could impact OL and myelination only at juveniles and the effect is transient. How do the authors explain it? Leaky Cre-deletion could not be the explanation. Also, it is unclear at what age between P14 and P60 the effect of Tet1 KO was lost. It is known that there is some technical issue with *Olig1-Cre* line as it also affects *Oligo2*.
3. The authors claimed that the conditional Tet1 KO mice displayed schizophrenia-like phenotype. The assays that they presented are typical assays used for modeling neurodevelopmental disorders and I don't believe they could claim they are schizophrenia-specific phenotypes.
4. Previous studies have shown the defect in Morris Water Maze assay for Tet1 KO mice. The authors should cite the original papers. Also, they should display the data using box plot.
5. In Figures 6-8, the authors attempted to identify the key downstream targets of Tet1 that could contribute to the observed phenotype. The analyses that the authors have performed are quite superficial and they should perform TET1 ChIP-seq to directly identify the downstream targets. The data that they have right now is cherry-picking and they need to provide a better explanation.
6. Figure 8 does not really address the effect of 5hmC on *Itpr2*. I would like to see the overexpression of *Itpr2* cDNA transgene could rescue Ca^{++} defects in TET1 cKO mice.
7. The authors should also compare and discuss the findings in reference #23.

Point-by-point Responses to Reviewers' concerns:

Reviewer #1 (Remarks to the Author):

Zhang et al. showed that a genome-wide shift in 5-hydroxy-methyl-cytosine (5hmC) landscape during NPC to OPC (so OL lineage specification) and hypothesized that such changes are caused by TET1, an abundant member of TET family in OPCs. They characterized Olig1-Cre TET1 cKO mice and revealed that TET1 functions are required for early OPC cell proliferation control and OL lineage progression, as well as remyelination. However, the cKO phenotypes were only prominent at young ages. This study further analyzed the potential targets of TET1 using RNA-seq and hMeDIP-seq approaches and proposed numerous calcium channels and intracellular transporters, such as type II IP3R. Interestingly, cKO OPCs exhibit altered intracellular calcium dynamics in response to some VOCC agonists. Also, the authors characterized Olig1-Cre-mediated *Itpr2* cKO, showing similar myelin defects found in *Tet1* cKO.

This study delivers important mechanistic insights into how DNA demethylation (specifically, TET1-dependent deoxygenation of methylcytosine) regulates early OL development and myelin repair, and their potential relevant mechanisms. However, given the importance of these scientific findings, the following points should be addressed, and the results interpreted accordingly.

We thank the reviewer for commenting on the significance and impact of our study to provide important mechanistic insights into how DNA demethylation regulates OL development and myelin repair.

1) *Page 3 - 4 in Results: This study compared 5hmC distribution between NPCs and OPCs in Fig.1. The results from this comparison should be interpreted as changes that occur during from NPC to OPC transition. The most appropriate term describing this process may be during 'OL lineage specification' as the authors used in introduction (page 3 line22). However, in this study, this is confused by mixing up terms 'OL lineage differentiation (page 3 line 35)', 'OL lineage transition (page 3, line 36)', or OL lineage progression (page 4, line 19), all of which potentially misleading or confusing future authors. The latter three refer to "OPC to mature oligodendrocyte transition".*

Response: Thank you for bringing this to our attention. We agree with the suggestion and have changed the descriptions to be more precise in our terminology (page 3 line 22, 35, 36; page 4 line 23).

2) *Page 4, line 2 - 3: Although the OPC isolation was described in methods section, it may help readers' understanding of the nature of OPCs by stating, "OPCs isolated via immunopanning from P6 mice".*

*Consider using genetic designations more consistently or differently: *Tet1^{f/f}* (page 4 line 25) or *Tet1^{flox/flox}* (line 27). *Olig1^{Cre} +/-* > *Olig1^{cre}/+**

Response: We have been more precise in describing our methods (page 4 line 3) and our terminology throughout.

3) *One of the most critical issues in this study is whether Olig1-Cre is reasonably specific to OL lineage. Given some previous publications (Jakovcevski et al., 2005; De Biase et al., 2011; Qi et al., 2016), Olig1 may also be expressed neural progenitors during embryonic stages. Provide low power fluorescence and high power confocal brain images of Olig1-Cre; R26-tdTomato (Ai14) or Olig1-Cre crossed with any widely-used cre*

reporters.

Response: Olig1 was identified as an oligodendrocyte lineage gene^{1,2}, which is expressed at the progenitor stages preceding PDGFRa⁺ OPC formation, such as early primitive OPCs. Although some studies referred to Olig1 expression in “neural progenitors”, they are likely primitive OPC progenitors since Olig1 and Olig2 are hardly detected in the general Nestin⁺ neural stem cells in the developing cortex. To address the specificity of Olig1-Cre expression, we have examined Olig1-Cre;R26-EYFP mice at P12 and find more than 95% of YFP⁺ cells are Olig2⁺ oligodendrocyte lineage cells in the brain (Revised Supplementary Fig. 3c-d). To address the concerns about the extent of recombination in neurons, we stained neuronal markers e.g. NeuN to show that there is no co-labeling of GFP with neuronal markers in the cortex (Revised Supplementary Fig. 3e).

4) Supplementary Figure 5 / Discussion

In discussion, the authors stated, “Tet1 cKO in the oligodendrocyte lineage exhibit impaired PPI and working memory deficits, the key behavior phenotypes related to schizophrenia. Thus, our studies demonstrate a crucial function of TET1-mediated epigenetic modifications in oligodendrocytes for schizophrenia and related brain disorders.”

- What is age of the mice used for this behavior test? It is hard to find the information of mice ages for the behavior test. If adult mice were used, how about OL and myelin phenotypes for those ages? Are these behavioral abnormalities related to myelin phenotypes in Tet1 cKO mice? At the ages of those mice, did authors see myelin abnormalities in cKO mice?

Response: For the PPI test, the mice were at P42, a juvenile stage, and the Tet1 mutant mice show abnormal myelination of white matter tracks (Revised Supplementary Fig. 5a). This result indicates that the impaired sensorimotor gating ability is correlated to myelin phenotypes in the Tet1 mutant. In the MWM test, where the mice were adults at P90, Tet1 mutant mice showed minimal myelination defects despite the initial delay in OL differentiation (Revised Supplementary Fig. 5d). Therefore, the behavioral abnormality may not be due to an acute myelin deficiency in Tet1 cKO mice at P90. We have provided the age information related to myelin abnormalities in cKO mice used for the behavior test (page 6 line 14 and line 24).

- In order to support the claim that oligodendrocyte (or OL lineage) -specific TET1 is critical PPI and working memory, it will be important to confirm that Olig1-Cre is really specific to OL lineage. Again, images from Olig1-Cre; R26-tdTomato (Ai14) for ventral brain, hippocampus and cortex should be provided. Otherwise, the interpretation must be more carefully stated and modified.

Response: Please see above response to question 3#. We have confirmed that Olig1-Cre is predominantly restricted to the OL lineage.

5) Page 7 lines 3-5, Fig.3 f–g:

Concerning the interpretation of the flow cytometry of PI-labeled OPCs: What is the block mechanism for G2/M to G1 transition? Is there any reference that described a block of G2/M to G1 transition? Alternatively, is there the possibility that this is caused by G2 arrest? It would be great if the authors use phospho-H3 (an M-phase marker) to distinguish G2 arrest from other mechanisms.

Response: Based on flow cytometry assays, we find a higher proportion of Tet1 cKO OPCs at the G2/M phase

of the cell cycle compared with control OPCs, suggesting that *Tet1* deletion impairs cell cycle progression at the G2-M phase in OPCs. This is in keeping with a recent report showing a critical role of TET1 in the G2 to M phase progression of trophoblast stem cells³. We have clarified the text by stating that *Tet1* deletion leads to G2/M transition rather than “a block of G2/M to G1” transition (page 15 line 16). We apologize for the confusion.

As suggested by the reviewer, we have examined G2-M phase progression in a cell cycle synchronization assay³ by immunostaining with phospho-H3, an M-phase marker⁴. After cell cycle release, we find that there is a delay in M-phase entry and withdrawal in *Tet1* cKO OPCs compared with control OPCs (Revised Fig.3h-j). This result suggests that *Tet1* deletion impairs cell cycle progression at the G2/M transition. The precise mechanism of the G2/M transition arrest in *Tet1* cKO OPCs is unknown, however we believe it is beyond the scope of the present study and it will be investigated in a future study.

Are the OPC cell cycle abnormalities seen at older ages? At least, the authors need to provide data of the density of BrdU+Olig2+ cells at different ages.

Response: As suggested by the reviewer, we have examined the density of BrdU+Olig2+ cells in older mice (P20 and P60) and did not find a significant difference in the density of BrdU+Olig2+ cells between control and *Tet1* cKO groups at older ages (Revised Supplementary Fig. 6c-f).

*6) How does *Tet1* deletion affect numbers of OPCs only up to P4? Is this short-effect related to compensation by other TET or TET1 expression declines with age (from P4 to P30)?*

Response: The exact mechanism restricting the effect of *Tet1* deletion on OPCs to the neonatal stage is unclear. This may in part be due to the substantial decline in TET1 expression with age particularly after P4⁵, suggesting a less critical role for TET1 in later stages of development. In addition, it is possible that compensation from other TETs (e.g. TET2 and TET3) might also contribute to OPC development in the *Tet1* cKO mice. We have discussed this point in the manuscript (page 7 line 8).

Can in vitro OPC culture (cKO and control) repeat similar results (i.e., cell cycle block)?

Response: The cell cycle analysis presented previously (original Figure 3f-g) was performed using a flow cytometric assay of cultured OPCs *in vitro*. In addition, we have repeated this experiment with the BrdU incorporation assay and confirmed the defect in cell cycle progression in *Tet1* cKO OPCs (Revised Supplementary Fig. 6g-i).

*7) Page 7 line 23: Correct the designation of *Tet1* iKO: NG2-CreERT: *Tet1*lox/lox (do not colon) > NG2-CreERTTM; *Tet1*lox/lox (CreERTTM and use semicolon)*

Response: We have corrected the designation of *Tet1* iKO as suggested (page 8 line 27).

8) The results of Fig.3 h and I (impaired OPC to OL transition in vitro) do not seem to fit the rest of Fig.3 (defective cell cycle). Shouldn't these results be moved to Fig.4? Can the decreased differentiation be quantified as the percentage of MBP+ cells/Olig2+ cells per a unit area?

Response: We thank the reviewer for the suggestion and have moved the results of Figs. 3H and I to Figure 4. In addition, we have also provided the quantification as the percentage of CNP⁺ or MBP⁺ cells/Olig2⁺ cells per unit area (Revised Fig. 4a-c).

9) *In order to test the possibility that the functional consequences of Tet1 deletion are compensated by other Tet members in Tet1 cKO (Fig. 2e), observing NG2-CreER Tet1 iKO at older ages will be highly significant. How about testing the same experiments of Figure 4 for adult ages (such as tamoxifen injection at P30 and observation at P70). The author should include Cre reporter to accurately assess the OPC fate and degree of OL lineage progression.*

Response: As suggested, we have performed tamoxifen-mediated deletion at P30 in NG2-CreER Tet1 iKO; R26-EYFP mice and analyzed the GFP reporter positive Tet1 iKO cells at P60. We did not detect any significant alteration in the OPC number and their differentiation into mature oligodendrocytes (CC1⁺) in Tet1 iKO mice (Revised Supplementary Fig. 8a-d), suggesting that Tet1 deletion does not alter OPC fate or differentiation.

10) *Unlike P1 brain of Tet1 cKO, Tet iKO exhibited that number of OPCs increased and OPC proliferation increased, indicating different patterns of OPC proliferation compared with P1 OPCs. Explain.*

Response: As Olig1-Cre mice can drive Tet1 ablation in the OL lineage from embryonic stages whereas NG2-CreER mice begin to delete Tet1 at the neonatal stages e.g., P2 after tamoxifen administration, the effect on the number of OPCs may be different at distinct developmental stages.

In Tet1 cKO mice, wherein Tet1 is deleted in the Olig1-Cre-expressing early precursors such as pri-OPCs⁶ from the embryonic stages, we observed a decrease in the number of OPCs, suggesting that TET1 regulates the transition of OPCs from the pri-OPC state. In Tet1 iKO mice, wherein NG2-CreER is expressed in the committed OPCs, we observed a defect in OPC differentiation and a corresponding increase in undifferentiated OPCs, suggesting that TET1 deletion in NG2⁺ OPCs impairs OPC differentiation into mature OLs and maintains them in the OPC state. Thus, TET1 regulates early commitment of OPCs from pri-OPCs and later differentiation of OPCs, producing stage-specific effects on OPC numbers.

We have discussed this point in the manuscript (page 8 line 35).

11) *Fig 5b -d: There are many Tet1+Olig2- cells after injury. What are those cells? The effectiveness of Tet1 deletion by NG2-CreER and the tamoxifen protocol?*

Response: As for TET1⁺Olig2⁻ cells in the lesion site, we have examined TET1 expression with cell type-specific markers (Nestin for neural precursor cells, GFAP for astrocytes, Iba-1 for microglia). We found that most Iba-1⁺ microglia and few Nestin⁺ cells were positive for TET1 (Revised Supplementary Fig. 8e), indicating that TET1 is also upregulated in microglia after injury.

We have also included the effectiveness of Tet1 deletion by NG2-CreER after injury, where approximately 97% Olig2⁺ cells lack TET1 expression at 7dpi, indicating an efficient deletion of Tet1 by NG2-CreER (Revised Fig. 5c-e). We have provided a detailed protocol for tamoxifen administration in the revised Results section (page 9 line 27) and Methods section (page 24 line13).

12) *This study did not address whether 5hmC levels changes in vivo, or genome-wide 5hmC landscape*

changes during OPC-to-OL transition (truly OL lineage progression). Can the authors address these question? Alternatively, the relevant discussion will be helpful to improve this manuscript.

Response: As suggested by the reviewer, we have assessed the genome-wide 5hmC landscape by hMeDIP-seq in OL and compared with that in the OPCs. Interestingly, we find increased 5hmC intensity in OL compared to OPC, which indicates the genome-wide 5hmC landscape changes during the OPC-OL transition. Considering the unique function of TET1 in oligodendrocyte lineage development, we speculate that the changes in 5hmC epigenetic marks may contribute to the OPC to OL transition. We have included these new data in the manuscript (**Revised Supplementary Fig. 10**).

13) In Fig.8 a-d: in order to separate Ca influx from Ca release-dependent calcium traces, can the authors use different agonists or Ca-free extracellular conditions in calcium imaging? What is the size of internal Ca store in cKO OPCs?

Response: To separate calcium influx from calcium release-dependent calcium traces, we have used a calcium-free extracellular condition, aCSF (artificial cerebrospinal fluid), to perform ATP-induced calcium imaging and detect calcium release from internal calcium stores. We find that *Tet1* deletion led to a defect in calcium release from internal calcium stores in cKO OPCs (**Revised Fig.9a-c**). In addition, we have examined the extracellular calcium influx induced by an L-VOCC agonist in calcium-containing aCSF. We find that calcium influx is reduced in *Tet1* cKO OPCs compared with control OPCs (**Revised Supplementary Fig. 11c-e**). Together, these data suggest that TET1 positively regulates both calcium influx and calcium release in OPCs.

To compare the size of internal Ca^{2+} stores between groups, we used mag-Fluo-4-AM, a low-affinity fluorescent Ca^{2+} indicator, to measure the relative Ca^{2+} level in endoplasmic reticulum as previously reported⁷. Living OPCs from both groups showed similar relative mag-Fluo4-AM fluorescence in the ER, as measured by spectrofluorometer (**Revised Supplementary Fig. 11a-b**).

*14) In Fig. 8i and j: IP3R (type II) is also known to be expressed in astrocytes. The authors should show astroglial *Itpr2* levels remain unaltered.*

Response: According to a neural cell-type transcriptome database⁸, *Itpr2* expression is lower in astrocytes compared to newly formed oligodendrocytes (**Revised Supplementary Fig. 12a**). As suggested, we have nevertheless performed qRT-PCR assay and showed that astroglial *Itpr2* mRNA expression remains constant in purified astrocytes from *Itpr2* cKO mice (**Revised Supplementary Fig. 12g**).

*15) In Fig. 8k – m: Can the authors add one or two more older ages regarding numbers of OPCs and OLs in control and *Itpr2* cKO?*

Response: We have assessed the number of OPCs and OLs in P7 and P14 *Itpr2* cKO mice, and showed that *Itpr2* cKO resulted in an OPC differentiation defect. We have now included an older age, P60, for analysis and find that there is no significant difference in the number of OPCs and OLs between control and mutant (**Revised Supplementary Fig. 12h, Revised Fig. 9k-l**), suggesting that the OL differentiation defect is recovered in adult *Itpr2* cKO mice.

16) *The authors stated in discussion. “Despite early developmental defects, we noticed that the developmental myelin deficiencies recovered in adult Tet1 cKO mice, which might be due to the expansion of OL numbers that escaped from Cre-mediated Tet1 depletion.” Can this be tested by TET1 co-immunostaining with CC1 at the adult ages.*

Response: As suggested, we have performed the co-immunostaining and detected low TET1 expression level in CC1⁺ OLs in the wildtype adult mice (Revised Supplementary Fig. 4f-g). In *Tet1* cKO adult mice, there is a population of CC1⁺ OLs expressing TET1 (Revised Supplementary Fig. 4f-g). Therefore, we believe that the expansion of OPCs that escaped from Cre-mediated *Tet1* depletion, from ~12% TET1⁺ in P4 *Tet1* mutants (Fig. 2b-c) to ~29% TET1⁺ in adult mutants, might contribute, at least in part, to recovery from a developmental myelin deficiency.

17) *In the title and abstracts: “TET1-mediated oligodendrocyte homeostasis ...” What do the authors mean by ‘oligodendrocyte homeostasis’? The meaning of homeostasis implies any intracellular mechanisms?*

Response: Homeostasis here refers to epigenetic landscape whereby the epigenetic state of oligodendrocyte lineage cells is maintained. To avoid the potential confusion, we have revised the title as “TET1-DNA hydroxymethylation mediated epigenetic landscape is required for CNS myelination and remyelination”.

Reviewer #2 (Remarks to the Author):

The manuscript by Zhang et al revealed TET1 mediated DNA hydroxymethylation is important and necessary for CNS Myelination and Remyelination. Their results showed the olig1-Cre mediated Tet1 depletion mice exhibit myelination deficits at the early postnatal stage, thus induce schizophrenia relevant behaviors, like impaired PPI and working memory; the underlined mechanism has also been experimentally explored as the OPC cell-cycle is blocked at the transition from G2/M to G1 phase in Tet1 cKO mice. The impaired remyelination phenotype in old mice brain has also been observed upon Tet1 loss, which is consistent with the recent report by Moyon et al. The author identifies several calcium transport related genes as major Tet1 targets which contribute to deficient OL differentiation, especially *Itpr2*, and ablation *Itpr2* in mice will lead to reduced OL development. Overall, this work illuminates a novel and unique role of TET1, but not TET2 or TET3, in OPC and OL differentiation, the underlined mechanism study is solid and clear, their findings are significant and meaningful.

We thank the reviewer for commenting the significance, soundness and importance of our study.

Following are my main comments:

1. According to the authors' previous report (Zhao et al, 2014), Zhao et al showed all TETs members are expressed in OLs and required for the differentiation of OLs *in vitro*. Here, the authors identified only TET1 as an impact factor in OL differentiation, could the authors give a proper explanation on the different results?

Response: The identification of TET1 as important factor for OL differentiation is consistent with the previous observation with siRNA-mediated knockdown in rat OPC cultures. However, in contrast to the *in vitro* data, we did not detect a significant effect of TET3 conditional knockout on OPC differentiation in the mouse model. The exact reason underlying the discrepancy is not clear. This might be due to possible off-target effects of siRNAs on other Tet members such as TET1 or other unidentified factors crucial for OPC differentiation. We have discussed this point in the manuscript (page 5 line 13).

2, Based on a recent report (Moyon et al, 2019), *Tet1* cKO has little effect on OPC to OL differentiation in the young mice brain (like P14), the two results (including CC1, Olig2, MBP staining) are inconsistent. How can the authors explain this, and which result is more reliable?

While Moyon et al reported several members of the solute carrier (*Slc*) gene family, including *Slc12a2*, are responsible for defective myelin regeneration detected in old *Tet1* cKO mice, so which one is more important? The solute carrier (*Slc*) gene family or calcium transport related genes, like *Itpr2*, reported by the authors in this study? The authors could compare the difference and discuss it. Are the solute carrier (*Slc*) gene family also found in the gene list of downregulated and low 5hmc enriched overlap in *Tet1* cKO mice in this study?

Response: According to the Moyon's preprint in BioRxiv, the authors indicate that *Tet1* cKO (*Tet1*^{loxP/loxP}; *Olig1-Cre*^{neo}) has little effect on OPC-to-OL differentiation in the young mouse brain. The reasons for different phenotypes between two mouse lines are not immediately clear. We speculate that the use of different *Tet1*-floxed strains might contribute to the phenotypic discrepancy during early developmental stages. In the present study, the *Tet1*-mutant mice lack exons 11-13, encoding the catalytic domain of TET1 enzyme⁹, while *Tet1*-mutant mice lacking exon 4 (*Tet1*Δe4)¹⁰ were used by Moyon et al. Distinct phenotypes and gene expression changes have been reported among individual *Tet1* mutant mice lacking exon 4¹⁰ or exon 11-12

⁹. Thus, it is possible that the phenotype discrepancy observed in different *Tet1*-mutant mice might be due to the nature of the mutations disrupting specific *Tet1* isoforms, as suggested by the alternative splicing forms of the *Tet1* gene ¹¹. In addition, other possibilities such as differences in *Olig1-Cre* lines (*Olig1-Cre* vs *Olig1-Cre^{neo}*), mosaicism and mouse backgrounds used in the studies might also contribute to the phenotypic discrepancy. We have discussed this point in the manuscript (page 14 line 27).

Similar to the Moyon's study, we have also detected *Slc* gene family members as TET1-5hmC targets, such as *Slc12a2*, *Slc24a2*, *Slc8a1*, and *Slc7a14* (original Fig 7h-j, and Revised Supplementary Fig. 9g), in addition to *Itpr2*. We show that *Itpr2* is highly enriched in the OPCs and newly formed oligodendrocytes (Fig 9d) ^{12, 13}. Moreover, our conditional knockout study in mice demonstrated that *Itpr2* is required for oligodendrocyte myelination. It is possible that TET1-mediated genome-wide DNA hydroxymethylation may regulate multiple distinct targets including *Itpr2* and *Slc12a2* for optimal myelinogenesis or at the different stages of oligodendrocyte lineage cells. It's possible that both genes are required for normal myelination, necessary but alone insufficient. We have discussed this point in the manuscript (page 17 line 20).

3, The authors identified more gene body regions associated with OL lineage progression when comparing the hMeDIP-seq data in OPC and NPC. Later, they also focus on genes that show both downregulated mRNA expression and decreased gene body 5hmC levels in Tet1 cKO mice OPC. Could the authors try to explain how gene body 5hmC affect gene expression? In their discussion, they mentioned histone H3K27me3/H3K4me3 modification may contribute to this, but this "5hmC accumulates in bivalent domains" mechanism has little relationship with the gene body regions and is not convincing.

Response: Thanks for the comments. The association of 5hmC enrichment in the gene body with active transcription has been reported in different tissues ^{14, 15, 16, 17, 18}, although the exact mechanism by which gene body 5hmC regulates gene expression is not clear at present. One possibility is that 5mC oxidation relieves a repressive effect on transcription, perhaps by counteracting spurious intragenic anti-sense transcription ¹⁹. Other explanations may include the fact that 5hmC has a destabilizing effect on DNA structure that potentially favors the opening of the double helix by the transcription apparatus ^{20, 21}. Moreover, 5hmC enriched in the gene body cannot be bound by transcriptionally repressive methyl-CpG binding domain (MBD) proteins *in vitro* ^{22, 23}. These studies suggested that gene-body 5hmC levels may regulate the transcription rate by modifying the accessibility of genic chromatin to transcriptional machinery or by inhibiting binding of repressive methyl-CpG binding proteins (MBDs).

We have discussed this point in the revised manuscript (page 16 line 20).

Besides, while the authors showed TET1 has dual effect on gene expression: activation or repression, why only the active genes are related to OL differentiation? The authors should try to illuminate them in the discussion.

Response: Our data indicate that, like many other epigenetic regulators, TET1 has a dual effect on gene expression: activation or repression depending on the gene loci, although the underlying mechanism is not fully defined. We found a set of TET1-activated genes that are related to OL differentiation. This may reflect a cell-type-specific transcriptional regulation for TET1 in oligodendrocyte differentiation. We have discussed this point in the manuscript (page 10 line 20).

4, While the authors show the ablation of Tet1 results in OPC cell-cycle progression, they performed flow

cytometry and found increasing percentages of cells in S and G2/M phase, but decreasing in G1 phase. To make their conclusion more persuasive, the authors could perform cell cycle synchronization assay to clearly and directly show the defects in OPC cell-cycle.

Besides, the authors show Ccng2, Cdca3, Ccna2, Ccnb2, and Cdc25b, which encode factors involved in cell-cycle regulation are enriched by 5hmc modification, may contribute to cell cycle blockade. While TET1 is also critical to the stability of cyclin B1 and acts as a facilitator of mitotic cell-cycle progression, which pathway is the major contribution to cell cycle inhibition: TET1-mediated 5hmc modification activates multiple gene expression or influence the cyclin B1 stability?

Response: Thanks for the suggestion. Please also refer to the response to Reviewer 1, point #5. We have performed experiments to examine G2-M phase progression using a cell cycle synchronization assay³ by immunostaining with an M-phase marker phospho-H3⁴. After cell cycle release, we find that there is a delay in M-phase entry and withdrawal in *Tet1* cKO OPCs compared with control OPCs (Revised Fig. 3h-j). This result suggests that *Tet1* deletion impairs cell cycle progression at the G2/M transition.

We have showed the downregulation of several cell cycle regulators such as *Ccng2*, *Cdca3*, *Ccna2*, *Ccnb1*, and *Cdc25b* in *Tet1* cKO OPCs (Figure 6h, Revised Supplementary Fig.6j-k). In particular, accumulation of *Ccnb1* (Cyclin B1) is a prerequisite for mitotic initiation in the late G2 phase and TET1 has been shown to be critical to the stability of cyclin B1 in trophoblast stem cells³. It is possible that TET1 may regulate the expression of multiple cell cycle genes, cyclin B1 stability, or both. However, we do not have any direct evidence showing TET1 regulates cyclin B1 stability. The precise mechanism for cell cycle transition block in *Tet1* cKO OPCs is unknown. Since our study focuses on the function of *Tet1* in oligodendrocyte development and myelination, we believe it is beyond the scope of the present study, and more appropriately investigated in a future study. We have discussed this point in the manuscript (page 15 line 19).

5, According to Supplementary Movie 1 and 2, before adding ATP and monitoring the [Ca²⁺] increase, the Tet1 cKO OPCs already have a very high fluorescence signal. The high background could cover the real calcium rise. The authors should repeat this assay using a similar initial fluorescence signal in control and Te1 cKO OPCs.

Response: We have repeated the assay and include the new data taking care to normalize the image parameters between the two conditions (Revised Fig. 8b-c, Revised Supplementary Movie 2).

6, While Itpr2 cKO mice indeed impair the calcium release and inhibit OL differentiation, these results couldn't demonstrate Itpr2 is a key downstream target of TET1-mediated 5hmc modifications which is responsible for impaired OL differentiation in Tet1 cKO mice. The authors should perform the rescue assay to put back Itpr2 in Tet1 cKO OL cultures to monitor the differentiation ability.

Response: We have attempted overexpressing *Itpr2* in *Tet1* cKO OPCs, however, the size of Mus *Itpr2* coding region was too large (8007 bp) to accommodate into an expression vector such as a lentiviral vector for its expression in OPCs. Instead, we used an *Itpr2* agonist, adenophostin A (Ada)²⁴, for the rescue experiments. Our data show that Ada addition significantly increases the differentiation ability of *Tet1* cKO OPCs (Revised Supplementary Fig. 12d-f). Together with the *Itpr2* cKO mice results, we believe that *Itpr2* at least in part mediates TET1 function for OPC differentiation.

7, In the discussion, the authors claim “Despite early developmental defects, we noticed that the developmental myelin deficiencies recovered in adult *Tet1* cKO mice, which might be due to the expansion of OL numbers that escaped from Cre-mediated *Tet1* depletion”, their explanation is not convincing, as the authors' previous report (Zhao et al, 2014), as well as a recent study (Moyon et al, 2019), show TET1 is significantly downregulated in old mice' OLs. The authors could easily detect the *Tet1* expression in adult *Tet1* cKO mice to verify this possibility, otherwise, the authors should try to give proper explanations.

Response: TET1 expression is downregulated in OLs from old mice, however, it is still detectable in CC1⁺ OLs in adult mice (Revised Supplementary Fig. 4f-g). We observed TET1 expression in a population of OLs (~29%) in adult *Tet1*-cKO mice (Revised Supplementary Fig. 4f-g). Thus, myelin deficiencies that recovered in adult *Tet1*-KO mice might be in part due to the escape from Cre-mediated *Tet1* depletion in a population of OL lineage cells. Similar observations of depletion escape have been observed in other studies, e.g. for the recovered myelin deficiency in adult *Dicer1* cKO mice ²⁵. In addition, it is also possible that the myelination process is not affected in adult OLs lacking TET1 expression, therefore, TET1 may not be essential for myelinogenesis in mature OLs. We have discussed this point in the manuscript (page 15 line 5).

Reviewer #3 (Remarks to the Author):

Ten-eleven translocation (TET) proteins-mediated cytosine hydroxylation has been shown to play important roles in neurodevelopment and diseases. However, their role in oligodendrocyte homeostasis, myelination and remyelination is unclear. In this manuscript, the authors provide the data suggesting that 5hmC and Tet proteins, particularly TET1, could play roles during oligodendrocyte-progenitor (OPC) differentiation. The topic is certainly interesting and important, however, the data presented in this manuscript are preliminary and lack molecular mechanistic insights. Below are some specific comments:

We thank the reviewer for appreciating the broad interest and importance of our study. Our study demonstrates a novel and unique role of TET1, but not TET2 or TET3, in OL development and myelin repair and further reveals the role of TET1-dependent hydroxymethylation in gene transcription during OPC development as well as a critical role for candidate TET1 target *Itpr2* in oligodendrocyte development. Consonant with comments by Reviewer 1# and 2#: "This study delivers important mechanistic insights into how DNA demethylation (specifically, TET1-dependent deoxygenation of methylcytosine) regulates early OL development and myelin repair, and their potentially relevant mechanisms". We believe that our study has provided important mechanistic insights into TET1-5hmC regulated OL lineage procession.

1. Figure 1-the authors used 5hmC antibody for IP followed by high-throughput sequencing to profile 5hmC dynamics during OL lineage differentiation. It is well known that 5hmC antibody IP-based approach has bias towards heavily methylated regions. The authors need to use some other methods to validate and confirm their findings.

Response: The 5hmC antibody IP-based approach has been widely used to profile 5hmC modifications in the genome ^{26, 27}, although it has a bias towards methylation-enriched sequences. However, since highly methylated CpG rich regions (e.g. CGI) are actively involved in transcriptional regulation, we leverage this method to investigate the distribution of 5hmC marks in OPCs and their potential functions in oligodendrocyte development. As the reviewer suggested, we have performed an unbiased loci-specific qPCR assay based on a combination of bisulfite and subsequent cytosine deaminase treatment (BisulPlus™ Loci 5mC & 5hmC Detection PCR Kit, EpiGentek, # P1067), to successfully validate some representative genes in Fig.1 (Revised Supplementary Fig.1d and Revised Supplementary Table 1). We have included the data in the manuscript.

2. Figure 2 and 3 suggest that the loss of TET1 could impact OL and myelination only at juveniles and the effect is transient. How do the authors explain it? Leaky Cre-deletion could not be the explanation.

Response: We have examined TET1 expression in oligodendrocyte lineage cells in adult *Tet1* cKO mice and are able to detect TET1 expression in a population of OLs (~29%), suggesting a potentially leaky Cre-mediated deletion or transgenic mosaicism (Revised Supplementary Fig. 4f-g). Similar observations of depletion escape have been suggested in other studies, e.g. for the recovered myelin deficiency in adult *Dicer1* cKO mice ²⁵. Nonetheless, it is also possible that the myelination process is not affected in adult OLs lacking TET1 expression. Therefore, TET1 may not be essential for myelinogenesis in a population of mature OLs. We have discussed the possibilities in the manuscript (page 15 line 5).

Also, it is unclear at what age between P14 and P60 the effect of Tet1 KO was lost. It is known that there is some technical issue with Olig1-Cre line as it also affects Oligo2.

Response: We have performed the quantification of CC1⁺ mature OLs in *Tet1* cKO mice at P21 and P27 (between P14 and P60), and found that the number of OLs in *Tet1* mutant corpus callosum is less at P21 and P27, but comparable at P60 (Fig 2e). As suggested by the reviewer, we have included additional stages e.g. P42 and P52 for analysis and found that the number of CC1⁺ mature OLs at P52 is comparable between control and *Tet1* cKO mice (Revised Fig. 2e and Revised Supplementary Fig. 4a). Therefore, we have more precisely defined the window over which the effect of *Tet1* KO on OL differentiation is lost to between P42 and P52.

In our studies, we used heterozygous Olig1-Cre (+/-) as the Cre driver and did not find any Olig2 expression changes with the Olig1-Cre (+/-) heterozygous line (Revised Supplementary Fig. 3f-g). Neither did Dr. Casaccia report any technical issue with the Olig1-Cre line in their study (reference #23).

3. The authors claimed that the conditional Tet1 KO mice displayed schizophrenia-like phenotype. The assays that they presented are typical assays used for modeling neurodevelopmental disorders and I don't believe they could claim they are schizophrenia-specific phenotypes.

Response: It is impossible to recapitulate the full phenotypic spectrum of schizophrenia in mice²⁸, researchers have proposed several core symptoms, e.g. hyperactive, stimulant sensitivity, social interaction deficits and working memory^{28, 29, 30, 31}. In particular, loss of normal prepulse inhibition of startle (PPI) is widely accepted as an endophenotype of schizophrenia and considered indicative of disrupted sensorimotor gating, a precognitive process to prevent sensory overload and cognitive fragmentation^{31, 32, 33}. Besides, the prodromal cognitive symptoms of schizophrenia often precede the occurrence of psychosis, and the range of cognitive deficits in schizophrenia suggests an overarching alteration in cognitive control, the ability to adjust thoughts or behaviors in order to achieve goals^{29, 34, 35}. Therefore, we choose PPI test and Morris water maze to examine the sensorimotor gating ability and working memory of *Tet1* cKO mice. In addition, recent studies suggest the schizophrenia as a neurodevelopmental disorder, including impaired synaptic connectivity, excitatory/inhibitory balance, and deficient myelination³⁶. Given the validity of any mouse model for such a complex disease as schizophrenia is not available, we have only phenomenologically described the behavior deficits in the *Tet1* cKO mice and discussed the phenotype in the discussion (page 16 line 2).

4. Previous studies have shown the defect in Morris Water Maze assay for Tet1 KO mice. The authors should cite the original papers. Also, they should display the data using box plot.

Response: We have cited the studies in the manuscript (page 15 line 31) and display the data using box plot (Revised Supplementary Fig. 5).

5. In Figures 6-8, the authors attempted to identify the key downstream targets of Tet1 that could contribute to the observed phenotype. The analyses that the authors have performed are quite superficial and they should perform TET1 ChIP-seq to directly identify the downstream targets.

Response: Given that the TET1 enzyme leads to 5hmC modifications in the DNA, we believe that the identification of 5hmC modifications in the genome by 5hmC high-throughput sequencing would provide

evidence for functional TET1 targets. Nonetheless, as requested by the reviewer, we have performed Cut&Run-seq (equivalent to ChIP-seq) to directly identify TET1 targets (Revised Fig. 8). Notably, TET1 binding profiles in representative genes were in accordance with 5hmC peaks, and associated with the activating histone mark H3K27ac, indicating a transcriptional activation role of TET1 in addition to its DNA methylcytosine dioxygenase activity in the regulation of gene transcription. These new data have been added in the manuscript (Revised Fig. 8).

The data that they have right now is cherry-picking and they need to provide a better explanation.

Response: To explore the mechanism whereby TET1 regulates OL differentiation, we have examined the RNA transcriptome and DNA hydroxymethylation alterations in OPCs after TET1 ablation. Such a screen is unbiased. The integrative analysis of RNA-seq and 5hmC profiling further reveal the candidate TET1-5hmC target genes, which are associated with OL differentiation, cell division, and calcium transport. TET1-Cut&Run assay for OPCs further indicates the involvement of TET1 regulation in certain group of targets.

While thorough analysis of all TET1 targets is beyond the scope of our study, we choose to analyze calcium transport-related genes since this pathway has not well characterized in oligodendrocyte development. We have confirmed the impaired calcium transport in *Tet1* cKO OPCs. Furthermore, we demonstrate a critical function for calcium transport *Itpr2*, as TET1-5hmC target, in OPC differentiation *in vitro* and *in vivo*, providing novel insight into potential mechanisms of TET1-5hmC regulated OL lineage procession. There is no reason to believe *a priori* that if we chose a different gene candidate from the RNAseq survey we would come to a different conclusion. We have provided an explanation for the gene selection for our study (page 12 line 9).

6. Figure 8 dose not really address the effect of 5hmC on Itpr2. I would like to see the overexpression of Itpr2 cDNA transgene could rescues Ca⁺⁺ defects in TET1 cKO mice.

Response: We have attempted overexpressing *Itpr2* in *Tet1* cKO OPCs, however, the size of Mus *Itpr2* coding region was too large (8007 bp) to package into a viral expression vector such as lentiviral vector for its expression in OPCs. Therefore, we used an *Itpr2* agonist, adenophostin A (Ada)²⁴, for the rescuing experiments. Our data show that Ada addition significantly increases the amplitude of ATP-induced calcium rise and the differentiation ability of *Tet1* cKO OPCs (Revised Supplementary Fig. 12d-f). Given that TET1 regulates a set of genes in the calcium transport, activation of a single gene *Itpr2* might not be able to fully rescue the calcium defects in *Tet1* cKO mice. Nonetheless, together with the 5hmC modifications on the *Itpr2* gene locus and *Itpr2* cKO phenotypes, our data suggest that *Itpr2* at least in part mediates TET1 function for OPC differentiation. We have discussed this point in the manuscript (page 13 line 31).

7. The authors should also compare and discuss the findings in reference #23.

Response: We have compared and discussed the similar findings in reference #23 posted in bioRxiv in the Discussion section in the manuscript (page 14 line 27).

Reference

1. Lu QR, *et al.* Sonic hedgehog--regulated oligodendrocyte lineage genes encoding bHLH proteins in the mammalian central nervous system. *Neuron* **25**, 317-329 (2000).
2. Zhou Q, Wang S, Anderson DJ. Identification of a novel family of oligodendrocyte lineage-specific basic helix-loop-helix transcription factors. *Neuron* **25**, 331-343 (2000).
3. Chrysanthou S, *et al.* A Critical Role of TET1/2 Proteins in Cell-Cycle Progression of Trophoblast Stem Cells. *Stem cell reports* **10**, 1355-1368 (2018).
4. Balistreri G, *et al.* Oncogenic Herpesvirus Utilizes Stress-Induced Cell Cycle Checkpoints for Efficient Lytic Replication. *PLoS pathogens* **12**, e1005424 (2016).
5. Zhao X, *et al.* Dynamics of ten-eleven translocation hydroxylase family proteins and 5-hydroxymethylcytosine in oligodendrocyte differentiation. *Glia* **62**, 914-926 (2014).
6. Weng Q, *et al.* Single-Cell Transcriptomics Uncovers Glial Progenitor Diversity and Cell Fate Determinants during Development and Gliomagenesis. *Cell Stem Cell* **24**, 707-723 e708 (2019).
7. Li N, Sul JY, Haydon PG. A calcium-induced calcium influx factor, nitric oxide, modulates the refilling of calcium stores in astrocytes. *The Journal of neuroscience : the official journal of the Society for Neuroscience* **23**, 10302-10310 (2003).
8. Zhang Y, *et al.* An RNA-sequencing transcriptome and splicing database of glia, neurons, and vascular cells of the cerebral cortex. *J Neurosci* **34**, 11929-11947 (2014).
9. Zhang RR, *et al.* Tet1 regulates adult hippocampal neurogenesis and cognition. *Cell stem cell* **13**, 237-245 (2013).
10. Towers AJ, *et al.* Epigenetic dysregulation of Oxtr in Tet1-deficient mice has implications for neuropsychiatric disorders. *JCI insight* **3**, (2018).
11. Zhang W, *et al.* Isoform Switch of TET1 Regulates DNA Demethylation and Mouse Development. *Molecular cell* **64**, 1062-1073 (2016).
12. Marques S, *et al.* Oligodendrocyte heterogeneity in the mouse juvenile and adult central nervous system. *Science* **352**, 1326-1329 (2016).
13. Zeisel A, *et al.* Brain structure. Cell types in the mouse cortex and hippocampus revealed by single-cell RNA-seq. *Science* **347**, 1138-1142 (2015).
14. Song CX, *et al.* Selective chemical labeling reveals the genome-wide distribution of 5-hydroxymethylcytosine. *Nature biotechnology* **29**, 68-72 (2011).
15. Nestor CE, *et al.* Tissue type is a major modifier of the 5-hydroxymethylcytosine content of human genes. *Genome research* **22**, 467-477 (2012).
16. Hahn MA, *et al.* Dynamics of 5-hydroxymethylcytosine and chromatin marks in Mammalian neurogenesis. *Cell reports* **3**, 291-300 (2013).
17. Jin SG, Wu X, Li AX, Pfeifer GP. Genomic mapping of 5-hydroxymethylcytosine in the human brain. *Nucleic acids research* **39**, 5015-5024 (2011).
18. Xu Y, *et al.* Genome-wide regulation of 5hmC, 5mC, and gene expression by Tet1 hydroxylase in mouse embryonic stem cells. *Molecular cell* **42**, 451-464 (2011).
19. Pfeifer GP, Kadam S, Jin SG. 5-hydroxymethylcytosine and its potential roles in development and cancer. *Epigenetics & chromatin* **6**, 10 (2013).
20. Wanunu M, *et al.* Discrimination of methylcytosine from hydroxymethylcytosine in DNA molecules. *Journal of the American Chemical Society* **133**, 486-492 (2011).
21. Lopez CM, Lloyd AJ, Leonard K, Wilkinson MJ. Differential effect of three base modifications on DNA thermostability revealed by high resolution melting. *Analytical chemistry* **84**, 7336-7342 (2012).
22. Valinluck V, Tsai HH, Rogstad DK, Burdzy A, Bird A, Sowers LC. Oxidative damage to methyl-CpG sequences inhibits

- the binding of the methyl-CpG binding domain (MBD) of methyl-CpG binding protein 2 (MeCP2). *Nucleic acids research* **32**, 4100-4108 (2004).
23. Jin SG, Kadam S, Pfeifer GP. Examination of the specificity of DNA methylation profiling techniques towards 5-methylcytosine and 5-hydroxymethylcytosine. *Nucleic acids research* **38**, e125 (2010).
 24. Ding Z, Rossi AM, Riley AM, Rahman T, Potter BV, Taylor CW. Binding of inositol 1,4,5-trisphosphate (IP3) and adenophostin A to the N-terminal region of the IP3 receptor: thermodynamic analysis using fluorescence polarization with a novel IP3 receptor ligand. *Molecular pharmacology* **77**, 995-1004 (2010).
 25. Dugas JC, *et al.* Dicer1 and miR-219 Are required for normal oligodendrocyte differentiation and myelination. *Neuron* **65**, 597-611 (2010).
 26. Ficiz G, *et al.* Dynamic regulation of 5-hydroxymethylcytosine in mouse ES cells and during differentiation. *Nature* **473**, 398-402 (2011).
 27. Gu TP, *et al.* The role of Tet3 DNA dioxygenase in epigenetic reprogramming by oocytes. *Nature* **477**, 606-610 (2011).
 28. Jones CA, Watson DJ, Fone KC. Animal models of schizophrenia. *British journal of pharmacology* **164**, 1162-1194 (2011).
 29. Marder SR, Fenton W. Measurement and Treatment Research to Improve Cognition in Schizophrenia: NIMH MATRICS initiative to support the development of agents for improving cognition in schizophrenia. *Schizophrenia research* **72**, 5-9 (2004).
 30. Winship IR, *et al.* An Overview of Animal Models Related to Schizophrenia. *Canadian journal of psychiatry Revue canadienne de psychiatrie* **64**, 5-17 (2019).
 31. van den Buuse M. Modeling the positive symptoms of schizophrenia in genetically modified mice: pharmacology and methodology aspects. *Schizophrenia bulletin* **36**, 246-270 (2010).
 32. Powell SB, Zhou X, Geyer MA. Prepulse inhibition and genetic mouse models of schizophrenia. *Behavioural brain research* **204**, 282-294 (2009).
 33. Mena A, Ruiz-Salas JC, Puentes A, Dorado I, Ruiz-Veguilla M, De la Casa LG. Reduced Prepulse Inhibition as a Biomarker of Schizophrenia. *Frontiers in behavioral neuroscience* **10**, 202 (2016).
 34. Mintz J, Kopelowicz A. CUtLASS confirms CATIE. *Archives of general psychiatry* **64**, 978; author reply 979-980 (2007).
 35. Lesh TA, Niendam TA, Minzenberg MJ, Carter CS. Cognitive control deficits in schizophrenia: mechanisms and meaning. *Neuropsychopharmacology : official publication of the American College of Neuropsychopharmacology* **36**, 316-338 (2011).
 36. Insel TR. Rethinking schizophrenia. *Nature* **468**, 187-193 (2010).

Reviewers' Comments:

Reviewer #1:

Remarks to the Author:

By the revision, the authors have addressed most concerns that I raised and improved the quality of the manuscript.

However, the cell type or lineage specificity of Olig1-Cre in the postnatal brain is still in question. In a paper (De Biase et al, 2011 J Neurosci 31(35): 12650–12662), it was shown that Olig1-Cre mice also express Z/EG reporter in cortical astrocytes (See Figure 4 A, B). The authors (Zhang et al) even commented on the possible difference between Olig1 Cre/+ and Olig1 neo, but any underlying scientific rationale for the difference is not clearly presented. Although the authors claimed that 95% of Olig1-Cre affected cells are Olig2+ OL lineage (in Supplementary Figure 3), the actual images do not agree well with the claim to my careful observation. My assessment of the provided images suggests that a much larger fraction of EYFP+ cells (than 5%) are not Olig2+. Also, it is not clear whether 5% of affected cells (non-Olig2+) by Olig1-Cre include interneurons (that are not always detected by NeuN staining) or astrocytes, both of which potentially impact the behavioral outcomes shown in supplementary Figure 5. The followings are suggestion:

1. Can the authors generate confocal images showing that EYFP+ (more brightly EGFP - immunostained images, not native EYFP) are not overlapped with ALDH111 or GFAP in CTX, hippocampus and corpus callosum?
2. In S. Figure 3e, it is clear that the EYFP cells are not overlapped with NeuN+ neurons (in the CTX). However, the EYFP+ perineuronal cells do not look like either OPCs or OLs. Can the authors show the same area images showing that EYFP+ are either NG2 OPCs or OLs, not being overlapped with ALDH111?
3. The results of Supplementary Figure 5 are interesting, but without more convincing reproducible results, it will be hard to believe that the memory or PPI phenotypes are only due to oligodendroglial Tet1 ablation. Thus, one alternative approach would be to use NG2-CreER or other PDGFRa-CreER (with tamoxifen) to reproduce similar results. If they can, the paper will be very strong, and the results can even be moved to the main figures.
4. Alternatively, the authors can delete S. Figure 5 results from this paper and use them for another separate scientific report with more supporting data.
5. In the current heading (for S. Figure 5), this is not "loss of Tet1 function in OLs", but "loss of TET1 function in OL lineage cells" because Olig1-Cre affects all OL lineage cells.

Reviewer #2:

Remarks to the Author:

The authors have address my concerns.

Reviewer #3:

Remarks to the Author:

The authors have addressed the majority of my concerns. However, I strongly suggest that the authors should remove "schizophrenia-like" description. All the assays that they used are related to neurodevelopment disorders and not schizophrenia specific. Claiming "schizophrenia-like" behaviors could be mis-leading.

Point-by-point Responses to Reviewers' concerns:

Reviewer #1 (Remarks to the Author):

By the revision, the authors have addressed most concerns that I raised and improved the quality of the manuscript. However, the cell type or lineage specificity of Olig1-Cre in the postnatal brain is still in question. In a paper (De Biase et al, 2011 J Neurosci 31(35): 12650–12662), it was shown that Olig1-Cre mice also express Z/EG reporter in cortical astrocytes (See Figure 4 A, B). The authors (Zhang et al) even commented on the possible difference between Olig1 Cre/+ and Olig1 neo, but any underlying scientific rationale for the difference is not clearly presented. Although the authors claimed that 95% of Olig1-Cre affected cells are Olig2+ OL lineage (in Supplementary Figure 3), the actual images do not agree well with the claim to my careful observation. My assessment of the provided images suggests that a much larger fraction of EYFP+ cells (than 5%) are not Olig2+. Also, it is not clear whether 5% of affected cells (non-Olig2+) by Olig1-Cre include interneurons (that are not always detected by NeuN staining) or astrocytes, both of which potentially impact the behavioral outcomes shown in supplementary Figure 5.

We thank the reviewer for the positive comments and careful assessment for our studies.

To further examine the cell-type or lineage-specificity of *Olig1-Cre* (without *neo* cassette) in the postnatal brain, we have generated *Olig1-Cre;R26-tdTomato* mice by crossing the *Olig1-Cre* mice with R26-tdTomato reporter line (Ai14), which expresses more robust tdTomato fluorescence relative to the EYFP reporter. Similarly, we found that most *Olig1-Cre-TdTomato* cells are Olig2-positive OL lineage cells (New Supplementary Fig. 3f-i), while TdTomato+ cells rarely co-label with the markers for astrocytes, neurons, or interneurons, which confirms that the *Olig1-Cre* line is predominantly restricted to the OL lineage. In addition, we validated that TET1 expression was not significantly decreased in neurons and astrocytes in the mutant brain (New Supplementary Fig. 7h). To exclude the potential impact of *Tet1* deletion in non-Olig2+ cells on the behavioral outcomes, as suggested, we have reproduced the PPI deficiency in mice with *NG2-CreER* mediated *Tet1* ablation (New Supplementary Figure 8a-b), indicating that oligodendroglial *Tet1* ablation can suffice to cause abnormal behavior phenotypes.

The followings are suggestion:

1). *Can the authors generate confocal images showing that EYFP+ (more brightly EGFP - immunostained images, not native EYFP) are not overlapped with ALDH111 or GFAP in CTX, hippocampus and corpus callosum?*

A: As suggested by the reviewer, we have examined tdTomato expression in Olig2 and ALDH1L1 labeled cells in the CTX, hippocampus and corpus callosum in *Olig1-Cre;R26-tdTomato* mice (Supplementary Fig. 3f-h). Quantifications indicate that most of the *Olig1-Cre-TdTomato* cells were Olig2+ OL lineage cells in all three brain regions (94.6% in CTX, 93.4% in Hippo, 92.2% in cc), which is consistent with previous observations (De Biase et al, 2011 J Neurosci 31(35): 12650–12662, Figure 4C. Both Figure 4A and 4B in the study did not show results for Z/EG reporter expression in cortical astrocytes as mentioned by the reviewer). We noticed that very few tdTomato+ cells overlapped with the astrocyte marker ALDH1L1 in these regions.

2). *In S. Figure 3e, it is clear that the EYFP cells are not overlapped with NeuN+ neurons (in the CTX). However, the EYFP+ perineuronal cells do not look like either OPCs or OLs. Can the authors show the same area images showing that EYFP+ are either NG2 OPCs or OLs, not being overlapped with ALDH111?*

A: We have provided the images showing that the majority tdTomato+ are in Olig2+ oligodendrocyte lineage cells and very few overlapping ALDH1L1 in the same area (New Supplementary Fig. 3f-h).

3). *The results of Supplementary Figure 5 are interesting, but without more convincing reproducible results, it will be hard to believe that the memory or PPI phenotypes are only due to oligodendroglial Tet1 ablation. Thus, one alternative approach would be to use NG2-CreER or other PDGFRa-CreER (with tamoxifen) to reproduce similar results. If they can, the paper will be very strong, and the results can even be moved to the main figures.*

A: We thank the reviewer for the suggestion. We have used the *Tet1^{flox/flox};NG2-CreER* (*Tet1* OPC-iKO) mice for the PPI test and reproduced the behavioral deficiency in *Tet1* iKO mice (New Supplementary Figure 8a-b). This observation indicates that *Tet1* ablation in OPCs can lead to abnormal behavioral phenotypes.

4). *Alternatively, the authors can delete S.Figure 5 results from this paper and use them for another separate scientific report with more supporting data.*

Please see the response to point 3#.

5). *In the current heading (for S. Figure 5), this is not “loss of Tet1 function in OLs”, but “loss of TET1 function in OL lineage cells” because Olig1-Cre affects all OL lineage cells.*

A: We agree with the reviewer and have modified the heading for Supplementary Figure 5 accordingly.

We thank the reviewer for the insightful suggestions.

Reviewer #2 (Remarks to the Author):

The authors have addressed my concerns.

We thank Reviewers 2 for helping us improve our manuscript.

Reviewer #3 (Remarks to the Author):

The authors have addressed the majority of my concerns. However, I strongly suggest that the authors should remove "schizophrenia-like" description. All the assays that they used are related to neurodevelopment disorders and not schizophrenia specific. Claiming "schizophrenia-like" behaviors could be mis-leading.

A: We thank the reviewer for the comments and have modified our statement in the revised manuscript as “behavioral deficiency” (page 2 line 50; page 3 line 90; page 6 line 185; page 7 line 215).

Reviewers' Comments:

Reviewer #1:

Remarks to the Author:

The authors have addressed my concerns. Now the manuscript is more convincing and provides highly significant insights. It will be an important work in the field of myelin biology.

Point-by-point Responses to Reviewers' Comments

Reviewer #1 (Remarks to the Author):

The authors have addressed my concerns. Now the manuscript is more convincing and provides highly significant insights. It will be an important work in the field of myelin biology.

We thank Reviewers 1 for helping us improve our manuscript.